# RATE-DISTORTION OPTIMIZATION FOR TRANSFORMER INFERENCE

## ABSTRACT

Transformers achieve superior performance on many tasks, but impose heavy compute and memory requirements during inference. This inference can be made more efficient by partitioning the process across multiple devices, which, in turn, requires compressing its intermediate representations. In this work, we introduce a principled rate-distortion-based framework for lossy compression that learns compact encodings that explicitly trade off bitrate against accuracy. Experiments on language benchmarks show that the proposed codec achieves substantial savings with improved accuracy in some cases, outperforming more complex baseline methods. We characterize and analyze the rate-distortion performance of transformers, offering a unified lens for understanding performance in representation coding. This formulation extends information-theoretic concepts to define the gap between rate and entropy, and derive some of its bounds. We further develop probably approximately correct (PAC)-style bounds for estimating this gap. For different architectures and tasks, we empirically demonstrate that their rates are driven by these bounds, adding to the explainability of the formulation.

## 1 INTRODUCTION

Transformer architectures (Vaswani et al., 2017) are predominant for many machine learning tasks. During inference, splitting the transformer architecture into manageable modules of contiguous layers allows for better horizontal scaling across multiple heterogeneous devices. The intermediate output produced by a module must then be efficiently transferred to the next. For example, a mobile device could perform part of the inference process, compress the intermediate representation, and then transmit it to a cloud server to complete the task. We propose a method (*codec*) to compress the intermediate representation produced by a transformer module. It is a *lossy* compression codec, where the target representation is optimized for a *rate* (e.g. bitrate) and a corresponding tolerable task error (*distortion*).

In auto-regressive models such as language models, there are different ways in which the inference process can be divided between two or more devices. The same module could be deployed on multiple devices to perform inference for different time frames of a response. In those situations, the *KV cache* of the module must also be transmitted. This work focuses on situations where a single device is responsible for the entire response of its assigned modules, so only the input and output representations of the module need to be transmitted.

Inherent to *learnable coding* is an *entropy model* that predicts and imposes a probability distribution on a target representation. The predicted distributions are used to code the target representation – generating a binary code and recovering the representation from such code. We propose an auto-regressive entropy model based on transformers. It generates a secondary representation (*hyperprior*) that must also be coded (Ballé et al., 2018). Focusing on language models, we benchmark the rate-distortion (RD) performance of our proposed method against other entropy models.

Due to the data processing inequality (Cover & Thomas, 2006), the entropy of the output representation of each subsequent module can only decrease. It is often expected in learnable coding that the rate should correlate with the entropy of the coded representation (Choi & Bajic, 2022). Our empirical results show that this is not the case in transformer-based models using our proposed entropy model, in which the rate-distortion performance decreases as we further process the input.

We explain this behavior through the theory of *usable information* (Xu et al., 2020), which takes into account the modeling power and computational constraints of an entropy model. We define and provide different bounds for the $\mathcal{V}$-entropy gap – the difference between rate and entropy – and provide a generalization error bound in terms of the *Rademacher complexity* (Shalev-Shwartz & Ben-David, 2014) of the target representation, and the *Lipschitz constant* of the entropy model. We estimate these bounds empirically to justify observations.

The bounds on the $\mathcal{V}$-entropy gap explain why, due to the complexity limitations of the entropy model, the rate often operates at orders of magnitude higher than the entropy of the target representation. Moreover, the bounds of its generalization error explain why an increase in the complexity of the entropy model does not necessarily translate into better performance. This behavior is similar to the bias-variance trade-off in learning theory. Finally, we show that the $\mathcal{V}$-entropy gap and its generalization error can also increase with certain notions of complexity of the target representation, which, unlike its entropy, can increase in deeper layers.

## 2 PREVIOUS WORK

Several techniques for efficient inference of transformer architectures have been studied in the context of language models (Xu et al., 2024). Such approaches include request batching and scheduling (Yu et al., 2022; Agrawal et al., 2024; Sheng et al., 2023; He & Zhai, 2024; Patel et al., 2024), parameter pruning (Zhang et al., 2024), model quantization (Lin et al., 2024), token pruning (Jiang et al., 2023a), and sparse attention mechanisms (Beltagy et al., 2020). The approaches in (Sheng et al., 2023; He & Zhai, 2024; Patel et al., 2024) share computation between multiple processes, but none consider optimizing the transferring state for rate-distortion performance.

In learnable image compression (Ballé et al., 2017; 2018; He et al., 2022; Zou et al., 2022; Jiang et al., 2025), an *analysis transform* maps an image to a target representation. An *entropy coder* uses the probability distribution estimates from the entropy model to generate a compressed representation of the target representation. Once transferred to a secondary device, the same entropy coder and entropy model are used to recover the target representation. A *synthesis transform* maps the target representation to an approximation of the image. The methods of *coding for machines* (Choi & Bajic, 2022; Harell et al., 2022) generalize to other tasks in addition to image reconstruction, where the output of the system is the prediction of a machine learning task. While these existing methods work with convolutional models, in this work we focus on transformers.

In learnable coding for machines, a *task codec* is trained using an *information bottleneck*, which is the penalization, during training, of the rate of the target representation, as provided by the entropy model. This rate reduction often results in the loss of some of the information necessary to perform a task, which increases the task distortion (error). The information bottleneck method (Tishby et al., 1999; Friedman et al., 2001) formalizes this approach and is viewed as a method for rate-distortion optimization (Cover & Thomas, 2006; Belghazi et al., 2018).

Diverse entropy models have been proposed for learnable coding. Seminal work employed convolutional neural networks (He et al., 2022; Jiang et al., 2023b). More recent work uses transformer architectures (Zou et al., 2022; Li et al., 2024) and state-space models (Qin et al., 2024; Zeng et al., 2025). These architectures generate a hyper-prior, which is an additional representation that must be coded and transmitted. This hyper-prior is used as *side information* for an auto-regressive entropy model of the target representation. The distribution imposed on the target representation is a multivariate normal distribution with a diagonal covariance matrix. The distribution imposed on the hyper-prior is fully-factorized and non-parametric (Ballé et al., 2018). In (la Fuente et al., 2024), a Fourier basis is proposed to model a fully-factorized probability distribution. Using fewer parameters, this approach is able to fit more complex distributions. In this work, we propose a relatively simple entropy model that relies more heavily on the hyper-prior. We show empirically that there is no justification for the extra complexity of entropy models often proposed in the literature.

The rate of an *optimal* task codec is lower-bounded by the entropy of the target random variable (Bajić, 2025). Although there are substantial rate reductions compared to the more rate-demanding input reconstruction tasks (de Andrade & Bajic, 2024; Harell et al., 2025), to achieve optimal task performance, the resulting rate in many learnable task codecs is often orders of magnitude higher than the entropy of the target random variable. We show that the ability of a learnable codec to

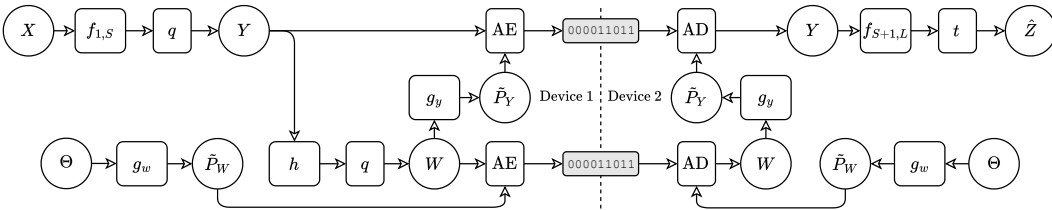

Figure 1: Architecture overview of the proposed codec. The AE and AD blocks correspond to arithmetic encoders and decoders, respectively. They use the probability distributions provided by the entropy models to encode their target representation into a bitstream and decode it back. The dotted line separates two devices, with bitstreams (gray blocks) connecting them.

achieve optimality is severely limited by the complexity of both the codec and its optimization. The lowest *achievable* entropy a set of probability functions $\mathcal{V}$ can measure in a random variable is formalized by the concept of $\mathcal{V}$-*entropy* (Xu et al., 2020). It can be estimated with guarantees if the complexity of $\mathcal{V}$ is bounded in terms of its *Radamacher complexity* (Shalev-Shwartz & Ben-David, 2014). It was shown in Xu et al. (2020) that bounds on the complexity of $\mathcal{V}$ directly translate to *probably approximately correct* (PAC) (Valiant, 1984) bounds for $\mathcal{V}$-entropy estimation.

## 2.1 PRELIMINARIES

This work focuses on decoder-only transformers (Liu et al., 2018; Radford & Narasimhan, 2018; Touvron et al., 2023) with a single coded intermediate representation, resulting in only two modules. It is straightforward to extend the proposed ideas to more modules and to encoder-decoder transformer architectures. Similarly, the proposed method could be applied to KV caches to support the distribution of the same module across multiple devices. This would allow the *compressed prefill-decode disaggregation* of language models (Zhong et al., 2024; Patel et al., 2024). To simplify notation without loss of generalization, all transformer layers produce representations of the same dimensionality as the input embeddings. Appendix A has a compilation of the notation used.

Assume $X \in \mathbb{R}^{T \times E}$ is an input random variable, where $T$ is the size of a time or spatial dimension, and $E$ is the embedding size. Let $\{f_l : \mathbb{R}^{T \times E} \to \mathbb{R}^{T \times E}\}_{l=1}^{L}$ be a set of $L$ transformer blocks describing the bulk of a transformer-based neural network. The first module of the network produces the target representation as $Y = (q \circ f_{1,S})(X); f_{1,S} = f_S \circ ... \circ f_1$, where $S$ is the *split point*, and $q$ is a *quantization function* (Agustsson & Theis, 2020; Theis et al., 2017; Ballé et al., 2016). This quantization (rounding) function discretizes the target representation so it can be coded. It has a differentiable training-time approximation that allows gradient propagation during automatic differentiation. The second module of the network produces the predictions as $\hat{Z} = (t \circ f_{S+1,L})(Y); f_{S+1,L} = f_L \circ ... \circ f_{S+1}$, where $t : \mathbb{R}^{T \times E} \to \mathcal{Z}$ are the header layers mapping to the sample space of the target random variable $Z$. A distortion function for $\hat{\mathbf{z}}, \mathbf{z} \sim (\hat{Z}, Z)$ is any task loss function $d(\hat{\mathbf{z}}, \mathbf{z})$ such that it is bounded and non-negative.

A hyper-prior (Ballé et al., 2018) is denoted as a random variable $W$ with sample space $\mathcal{W} \subseteq \mathbb{Z}^{T \times C}$. Let $\mathcal{F}(\mathbb{R})$ be the set of all *cumulative density functions* (CDFs), and $\Theta = \{\boldsymbol{\theta}_1, ..., \boldsymbol{\theta}_C\}$ be a set of parameter vectors for each embedding dimension of $W$. A zero-context learnable entropy model for the hyper-prior $g_w : \Theta \to \mathcal{F}(\mathbb{R})$ takes a parameter vector $\boldsymbol{\theta}_j; j = \{1, ..., C\}$ to generate a CDF for any element in the $j$-th embedding dimension of $W$. The rate of a hyper-prior $\mathbf{w} \sim W$ is the fully-factorized negative log-likelihood of a unit interval centered around $\mathbf{w}$:

$$r_w(\mathbf{w}) = -\sum_{i=1}^{T} \sum_{j=1}^{C} \log \left[ g_w[\boldsymbol{\theta}_j] \left( w_{i,j} + \frac{1}{2} \right) - g_w[\boldsymbol{\theta}_j] \left( w_{i,j} - \frac{1}{2} \right) \right]. \quad (1)$$

Let $\Omega = \{\mathcal{W} \cup \{\oslash\} \to \mathcal{P}(\mathcal{Y})\}$ be the set of functions that maps the side information $W$ (e.g.: hyper-prior) or a constant $\oslash$ to any probability distribution over the sample space of $Y$. A predictive family (Xu et al., 2020) $\mathcal{V} \subseteq \Omega$ is the set of predictive models a learning algorithm is constrained to use due to model limitations, optimization challenges, or *secondary objectives such as rate constraints*.

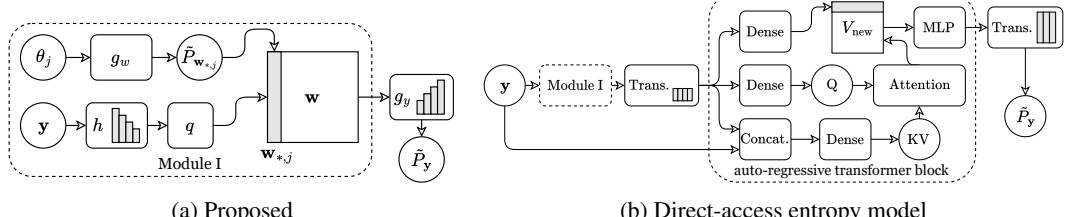

(a) Proposed                    (b) Direct-access entropy model

Figure 2: Architecture diagram of the different entropy models for the target representation $Y$. The direct-access entropy model replaces the proposed $g_y$ with a series of transformer blocks that combine the hyper-prior and the target representation. Q, K, V corresponds to the query, key, and value embeddings in an attention mechanism. The Fourier basis method is (a) with a different $g_w$.

The predictive conditional $\mathcal{V}$-entropy is defined as:

$$H_{\mathcal{V}}(Y|W) = \inf_{g \in \mathcal{V}} \mathbb{E}_{\mathbf{w}, \mathbf{y} \sim W, Y} \left[ -\log g[\mathbf{w}](\mathbf{y}) \right]. \tag{2}$$

Setting $\mathcal{V} = \Omega$ recovers the conditional entropy measure. In addition, setting $\mathcal{W} = \emptyset$ recovers Shannon's entropy.

## 3 CONTRIBUTIONS

### 3.1 AN AUTO-REGRESSIVE HYPER-PRIOR IS ALL YOU NEED

Although the functions generating the hyper-prior $W$ have access to the entire target representation $Y$, a rate constraint placed on $W$ reduces the information related to it. Hyper-priors are often used as side information by complex auto-regressive entropy models that also have access to previously coded elements in the target representation $Y$ (Ballé et al., 2018; He et al., 2022; Zou et al., 2022). This property alleviates the need for side information and can be insignificant in some cases. Hence, we propose a transformer-based entropy model that does not have *direct access* to elements in $Y$, requiring more support from the hyper-prior.

This design choice allows to code and transmit all representation elements within a time frame in parallel. It simplifies the entropy model of the target representation by assuming full independence of its elements given the hyper-prior. In Section 4.2, we compare our proposed method against a baseline that does not make this design choice, showing relatively better performance. The theory developed in Section 3.3 shows that the complexity of the entropy model increases the generalization error of the $\mathcal{V}$-entropy gap and this can negatively affect the rate-distortion performance of a codec.

Thus, our learnable entropy model $g_y : \mathcal{W} \rightarrow \mathcal{N}(\mathcal{Y}); \mathcal{W} \subseteq \mathbb{Z}^{T \times C}, \mathcal{Y} \subseteq \mathbb{Z}^{T \times E}$ estimates the parameters of a fully-factorized multivariate normal distribution assumed for $Y$ using $W$ as the *only* context (without direct access to $Y$). Assuming a conditionally independent distribution for $Y$ given $W$, the rate for $\mathbf{y} \sim Y$ is given by the negative log-likelihood of a unit interval centered around $\mathbf{y}$:

$$r_y(\mathbf{y}; \mathbf{w}) = - \sum_{i=1}^{T \times E} \log \left[ \Phi \left( y_i + 1/2; \ g_\mu(\mathbf{w})_i, \ g_\sigma(\mathbf{w})_i \right) - \Phi \left( y_i - 1/2; \ g_\mu(\mathbf{w})_i, \ g_\sigma(\mathbf{w})_i \right) \right], \tag{3}$$

where $g_\mu : \mathcal{W} \rightarrow \mathbb{R}^{T \times E}$ and $g_\sigma : \mathcal{W} \rightarrow \mathbb{R}_+^{T \times E}$ correspond to the means and variances, respectively, produced by the entropy model $g_y$, $\Phi$ is the normal CDF, and $i$ indexes the elements in the tensors.

A transformer-based hyper-prior model is given by $W = (q \circ h)(Y); h : \mathcal{Y} \rightarrow \mathbb{R}^{T \times C}$. It is designed to enforce a dimensionality bottleneck, with each transformer block gradually decreasing the dimensions of the representation to $T \times C$. The transformer in $g_y$ gradually increases the dimensionality to $2 \times T \times E$, where the first dimension is represented by $g_\mu$ and $g_\sigma$. The change of dimensionality is done by a projection layer injected between the attention layer and the *multi-layer perceptron* (MLP) sub-block of the transformer block. Figure 1 shows a diagram of the entire codec.

We propose a *deep factorized* density model (Ballé et al., 2018) as an entropy model for the hyper-prior $g_w$. It is a MLP with a single sigmoid output parameterized by $\boldsymbol{\theta}_j : j \in \{1, ..., C\}$, which

contains its weights, biases, and scaling factors for the activation functions. A monotonicity constraint is placed on $g_w$ to ensure it produces valid CDFs. This is achieved by a reparameterization of $\boldsymbol{\theta}_j$ to ensure that the function derivatives, in this case probabilities, are always non-negative. Figure 2a shows a diagram of the proposed entropy model.

The masks for the attention mechanisms in $h$ and $g_y$ are restricted to create representations that only depend on the current and previously coded elements. Although this restriction is not required to code the hyper-prior or the target representation, it allows to grow the existing side information by only appending elements to it as the time series is further processed. This is a critical feature for auto-regressive tasks that avoids the transmission of an entire hyper-prior for the inference of a new time frame; only the new elements generated by the new time frame need to be transmitted. The attention masks ensure that:

$$\tilde{P}(\mathbf{w}|\mathbf{y}) = \prod_{i=1}^{T} \prod_{j=1}^{C} \tilde{P}\left(w_{i,j}|\mathbf{y}_{\leq i}\right), \qquad \tilde{P}(\mathbf{y}|\mathbf{w}) = \prod_{i=1}^{T} \prod_{j=1}^{E} \tilde{P}\left(y_{i,j}|\mathbf{w}_{\leq i}\right), \qquad (4)$$

where $\mathbf{w}_{\leq i}$ and $\mathbf{y}_{\leq i}$ correspond to the elements in these tensors from current and previous time steps, and $\tilde{P}$ is a probability estimate implicitly established by the entropy models. As previously mentioned, another benefit of this property is that all elements within a time frame can be coded and transmitted in parallel. This has the potential to substantially reduce the inference latency that is due to coding.

Including the hyper-prior rate, the rate-distortion loss function is given by:

$$\mathcal{L}(Y, W, \hat{Z}, Z) = \mathbb{E}_{\mathbf{y},\mathbf{w},\hat{\mathbf{z}},\mathbf{z}\sim Y,W,\hat{Z},Z} \left\{ d(\hat{\mathbf{z}}, \mathbf{z}) + \lambda \left[ r_y(\mathbf{y};\mathbf{w}) + r_w(\mathbf{w}) \right] \right\}, \qquad (5)$$

where $\lambda \in \mathbb{R}_+$ balances the trade-off between rate (compression) and distortion (error). The two rates are affected equally by $\lambda$.

### 3.2 THE $\mathcal{V}$-ENTROPY GAP AND ITS BOUNDS

We extend the theory of *usable information under computational constraints* (Xu et al., 2020) to provide the $\mathcal{V}$-entropy gap, a measure that captures the limitations of an entropy model to show how a rate can operate at orders of magnitude higher than the entropy of a target random variable. We provide its connection to the rate-distortion loss function along several bounds for this measure.

**Definition 1.** *We define the $\mathcal{V}$-entropy gap as the absolute difference between the entropy and the rate of a random variable measured by the predictive family $\mathcal{V}$, with both terms expressed as conditional $\mathcal{V}$-entropies with optional side information $W$:*

$$G_{\mathcal{V}}(Y|W) \triangleq |H_{\mathcal{V}}(Y|W) - H_{\Omega}(Y|W)|. \qquad (6)$$

When the side information $W$ is $\oslash$ the gap can be expressed in terms of $\mathcal{V}$-entropies. The entropy gap can be also interpreted as the lowest expectation of KL divergences that a predictive family can produce. See Lemma 1 in Appendix A for details.

Assuming that the target representation $Y$ is a function of $X$, and that the side information $W$ is a function of the target representation $Y$, as it is the case for the hyper-prior, the $\mathcal{V}$-entropy gap is the lowest expected rate of the target representation achieved by the predictive family:

**Theorem 1.** *Let $\mathcal{V} \subseteq \Omega$ be a predictive family according to Xu et al. (2020), and let $Y = f(X)$, where $X$ is a random variable and $f$ is differentiable. Then:*

$$G_{\mathcal{V}}(Y|Y) = \inf_{g \in \mathcal{V}} \mathbb{E}_{\mathbf{x}\sim X} \left\{ -\log \left( g\left[f(\mathbf{x})\right] \circ f \right)(\mathbf{x}) \right\}. \qquad (7)$$

*Proof.* See Appendix A.

To adapt this result to our proposed method, we restrict the predictive family to produce multivariate normal distributions with diagonal covariances, such that $\mathcal{V} \subseteq \{\mathcal{Y} \cup \{\oslash\} \to \mathcal{N}(\mathcal{Y}; \Lambda)\}$, where $\Lambda$ is a set of all diagonal covariance matrices. Since in $G_{\mathcal{V}}(Y|Y)$, the side information is the target itself, the predictive family $\mathcal{V}$ must subsume the hyper-prior model $h$ together with the entropy model $g_y$ to meet this definition. Under these assumptions, for our definition of $Y$, we obtain:

$$G_{\mathcal{V}}(Y|Y) = \inf_{r \in \{v|v(\mathbf{y})=-\log g[\mathbf{y}](\mathbf{y}), g\in\mathcal{V}\}} \mathbb{E}_{\mathbf{x}\sim X} \left[ (r \circ q \circ f_{1,S})(\mathbf{x}) \right]. \qquad (8)$$

It is trivial to show that this result is the standalone minimization of the term $\mathbb{E}[r_y(Y; W)]$ in the loss function (Eq. 5). Thus, we can interpret part of the rate-distortion optimization problem as directly minimizing the $\mathcal{V}$-entropy gap. The rate restriction placed on $W$, in addition to the distortion penalty on $Y$ not carrying task-relevant information, and the potential limitations of the predictive family $\mathcal{V}$, prevent this $\mathcal{V}$-entropy gap from trivially reaching zero.

When the side information stems from the target representation, the $\mathcal{V}$-entropy gap is upper-bounded by the covariance determinant of the target representation. Theorem 4 in Appendix A.3 shows this for continuous variables. For the proposed method, we are interested in quantized target representations, such as $Y$:

**Theorem 2.** *Let $\mathcal{V} \subseteq \Omega$ be a predictive family according to Xu et al. (2020), and $Y = q(Y'; \Delta)$ a quantized continuous random variable where $\mathrm{cov}(Y') = \Sigma$, and $\Delta$ is the maximum quantization step. Then, with $D = |Y| = T \times E$, we have:*

$$G_{\mathcal{V}}(Y|Y) \leq {}^{1}\!/\!{}_{2} \log \det \Sigma + {}^{D}\!/\!{}_{2} \log (2\pi\mathrm{e}) + \inf_{g \in \mathcal{V}} \mathrm{KL}\left(P_Y \| g[\mathbf{y}]\right) - \log \Delta \quad as \; \Delta \to 0. \quad (9)$$

*Proof.* See Appendix A.

Many quantizers used in learnable coding have a fixed quantization step $\Delta = 1$, which effectively rounds values to their closest integer. With $\Delta = 1$, the provided bound is an approximation. However, the relationship between Shannon's entropy and differential entropy used for this bound is commonly used empirically, for relatively large values of $\Delta$, with compelling results (Cover & Thomas, 2006). If the target representation was originally a discrete random variable, there would be no impact from their covariance on the $\mathcal{V}$-entropy gap.

In Section 4.4, we measure an approximation of the covariance determinant of the target representations of different neural network layers. This quantity, found in Theorem 2, is independent of the capacity of the entropy model, which remains fixed across layers in the experiments, and is captured by the KL-divergence term also present in Theorem 2. We show a positive correlation between this measure and the rate, providing one reason why deeper layers can exhibit higher rate.

We derived other bounds on the $\mathcal{V}$-entropy gap showing interesting relationships between the Lipschitz constants of the functions involved. If we generalize to side information that stems from a common ancestor with the target random variable, we have the bound in Theorem 5 of Appendix A. If we exclude the hyper-prior function from the predictive family and keep it a function of the target representation $Y$, we have the bound in Theorem 6 of Appendix A.

### 3.3 GENERALIZATION BOUNDS FOR THE $\mathcal{V}$-ENTROPY GAP

In practice, we often use a set of samples to minimize the $\mathcal{V}$-entropy gap, since the joint distribution of the target representation and side information $P_{Y,W}$ is not known. Learning theory (Shalev-Shwartz & Ben-David, 2014) establishes the generalization error to be the hypothesis (e.g.: model parameters) producing the largest error between the expectation of its outputs and a corresponding empirical measure. This quantity can usually be bounded by the complexity of the hypothesis family (model architecture). Intuitively, a hypothesis family with smaller complexity is easier to learn.

Analog results exist for $\mathcal{V}$-entropy, where the complexity of the predictive family $\mathcal{V}$ result in probably approximately correct (PAC) (Shalev-Shwartz & Ben-David, 2014) bounds for $\mathcal{V}$-entropy estimation. We define the generalization error for the $\mathcal{V}$-entropy gap and establish similar bounds.

**Definition 2.** *Let $\mathcal{D} = \{(\mathbf{y}_i, \mathbf{w}_i)\}_{i=1}^{N} \sim Y, W$ be a set of samples. We define the generalization error of the $\mathcal{V}$-entropy gap as:*

$$R_{\mathcal{V},\mathcal{D}}(Y|W) \triangleq \left| G_{\mathcal{V}}(Y|W) - \inf_{g \in \mathcal{V}} \frac{1}{N} \sum_{(\mathbf{y},\mathbf{w}) \in \mathcal{D}} \log \frac{P_{Y|W}(\mathbf{y}|\mathbf{w})}{g[\mathbf{w}](\mathbf{y})} \right|. \quad (10)$$

This term can be upper-bounded in terms of the Rademacher complexity (Shalev-Shwartz & Ben-David, 2014) of the target representation and the Lipschitz constant of the predictive family $\mathcal{V}$.

**Theorem 3.** *Let $\mathcal{V} \subseteq \Omega$ be a predictive family according to Xu et al. (2020), $Y$ a random variable with sample space $\mathcal{Y}$, and $\mathcal{D} = \{\mathbf{y}_i\}_{i=1}^{N} \sim Y$ a set of samples. Assume that $\forall g \in \mathcal{V}, \mathbf{y} \in \mathcal{Y}, \log g[\mathbf{y}](\mathbf{y}) \in [-B, B]$. Then, $\forall \delta \in (0, 1)$, with probability at least $1 - \delta$, we have:*

$$R_{\mathcal{V},\mathcal{D}}(Y|Y) \leq 2 \operatorname{Lip}(\mathcal{V}_r) \operatorname{Rad}(\mathcal{D}) + B\sqrt{{}^{2}\!/\!{}_{N} \log {}^{1}\!/\!{}_{\delta}}. \quad (11)$$

*where* Rad *is the Rademacher complexity of a set, and* $\text{Lip}(\mathcal{V}_r)$ *is the maximum Lipschitz constant in* $\mathcal{V}_r = \{v|v(\mathbf{y}) = \log g[\mathbf{y}](\mathbf{y}), g \in \mathcal{V}\}$.

*Proof.* See Appendix A.

This result shows that the generalization error of the $\mathcal{V}$-entropy gap is upper-bounded by the Rademacher complexity of a set of target representations. As the set $\mathcal{D}$ grows in size (samples), the Rademacher complexity and the second term decrease, minimizing the generalization error. A more general bound, using a different random variable than $Y$ as side information, is presented in Theorem 7 of Appendix A.

In Section 4.4, we measure an approximation of the Rademacher complexity of the target representations of different neural network layers. This quantity is present in Theorem 3, along with other terms that remain constant across layers. We show a positive correlation between this quantity and the rate achieved on the target representations. This provides further reasoning as to why the rate might not correlate with the entropy of the target representations.

## 4 EXPERIMENTAL RESULTS

We evaluate the rate-distortion performance of the proposed architecture on language modeling. The results are compared against a similar architecture that uses Fourier basis functions (la Fuente et al., 2024) for the density estimation of the hyper-prior. In addition, we compare against the inclusion of a *direct-access* auto-regressive entropy model that has access to previously coded elements of the target representation and not just the hyper-prior, which contains only a subset of the information in the target representation. Figure 2 shows an overview of the two entropy models.

We train codecs with different values of $\lambda$ at multiple split points to obtain rate-distortion curves. The results show that under similar distortion performance, the $\mathcal{V}$-entropy gap increases with the split point. We attribute this behavior to an increase in the $\mathcal{V}$-entropy gap as well as its generalization error, demonstrated by increases in the covariance determinant and Rademacher complexity of the target representation through the network. This behavior is evaluated for other architectures and modalities, such as residual neural networks and images. All results are reported on validation sets. Anonymized code is available at anonymous.4open.science/r/lm-codec.

### 4.1 CONFIGURATION FOR THE PROPOSED ARCHITECTURE

The hyper-prior model $h$ is composed of 4 transformer blocks that sequentially bring down the embedding space $E$ to 384, 192, 96, and finally, $C = 24$ dimensions. Following (Karpathy, 2022), the dense layers in the transformer blocks have no biases, and the different sub-components have residual connections and layer normalizations. The deep factorized entropy model $g_w$ uses 9 dense layers with 3 hidden dimensions, for a total of $|\boldsymbol{\theta}_j| = 118$ parameters per $C$ dimension. The entropy models for the target representation $g_y$ is composed of another 4 transformer blocks, that sequentially bring up the embedding space to 96, 192, 384, and finally, $E$ dimensions. To quantize $Y$ and $W$, $q$ is set as a conventional integer rounding operation, where, during automatic differentiation, the incoming gradients are passed to the next operation as they were (Theis et al., 2017).

### 4.2 RATE-DISTORTION PERFORMANCE BENCHMARKS

We use the GPT-2 Small (Solaiman et al., 2019) transformer architecture as a language model. It has $L = 12$ transformer blocks with 12 attention heads and embeddings of $E = 768$ dimensions, resulting in 124 million parameters. As an auto-regressive language task, we use the OpenWebText (Gokaslan et al., 2019) dataset, mainly comprised of Reddit conversation threads.

The loss function is given by Eq. 5. Different values of $\lambda$ yield different points on the rate-distortion curve. The 6th transformer block was chosen as the split point. See Appendix B for more details and other results, including a detailed description of the comparative methods previously introduced.

Figure 3a shows the rate-distortion curves obtained for the proposed model and the baselines. The rate reported is the sum of the individual rates of the hyper-prior and target representations. We obtain a BD-rate (Bjontegaard, 2001), with respect the proposed architecture, of 99.46% and 10.7%,

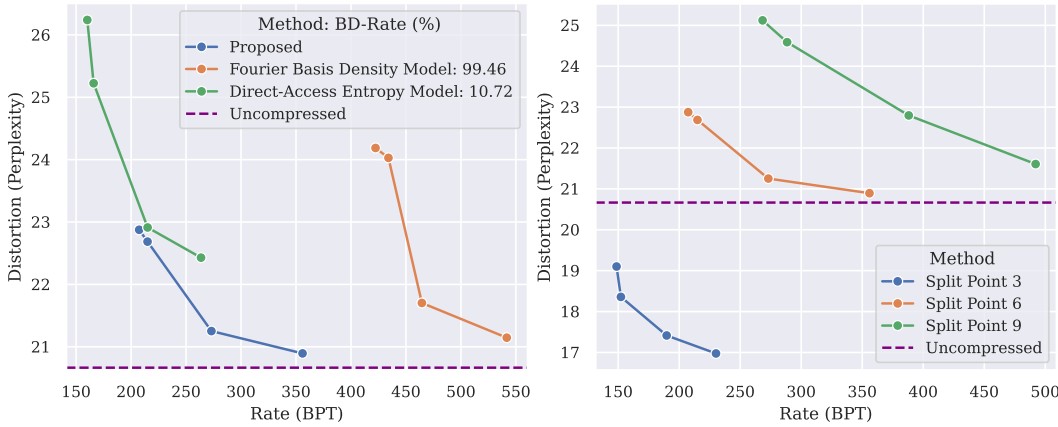

(a) Split point 6 under different entropy models    (b) Split points 3, 6, and 9 using the proposed method

Figure 3: Rate-distortion performance for GPT-2. The rate is measured in bits-per-token (BPT). Perplexity is the exponent of the classification cross-entropy loss, used as distortion. *Uncompressed* is a model with no quantization or rate penalty ($\lambda = 0$). The proposed method outperforms the other entropy models. Its rate-distortion performance decreases with the split point.

for the Fourier basis and the direct-access entropy model methods, respectively. Thus, our simpler architecture outperforms both baselines. Interestingly, the Fourier basis method uses much more bits for the hyper-prior.

### 4.3 BITRATE AND SPEED COMPARISONS

We use 16-bit precision on all language models. Assuming 16 bits per element in the target representation $Y$, our model from Section 4.2 with the lowest perplexity (highest bitrate for split point $S = 3$) uses 1.87% of the bits used by an uncompressed method.

We compare against Deflate (Deutsch, 1996), which is a lossless data compression file format commonly used for tensors. See Appendix C for more details and benchmark settings. For our proposed method, the inference on the entropy models used a GPU but the rest of the coding algorithm ran on a CPU. Deflate produces a bitrate of 1,053.88 BPT when compressing the target representations of our lowest perplexity model. Compared against the 230.16 BPT produced by this model, we operate at 21.84% of that bitrate. Deflate operates at 0.672 milliseconds per token, whereas our proposed method operates at 0.348 milliseconds per token. Our non-optimized method is 48.21% faster. If Deflate is used on the uncompressed method at split point $S = 3$, we obtain a bitrate of 11,415.13 BPT. Our proposed method operates at 2.02% of that bitrate. This result shows that even if we opt to use an off-the-shelf codec, there is still a significant benefit in using the representations induced by our proposed method.

We also compare against Zstandard (Collet & Kucherawy, 2018), which uses similar techniques as the DEFLATE algorithm, producing similar rates, but considerably faster. Zstandard achieves a bitrate of 1,172.16 BPT when compressing the target representations of our lowest perplexity model. Compared against our proposed method on the same model, we operate at 19.64% of that bitrate. If Zstandard is used on the uncompressed method at split point $S = 3$, we obtain a bitate of 11,383.07 BPT, which means our proposed method operates at 2.02% of that bitrate. Comparing speeds, Zstandard performs at 0.022 milliseconds per token, which is an order of magnitude faster than our method. However, we believe that our current implementation can be substantially optimized.

Compared to an uncompressed method where the raw tensor is transmitted, at our current non-optimized coding speed, and assuming a communication protocol overhead of 9% (Cavanaugh, 1994), our method is more efficient when the effective link speed is less than 37.77 Mbps.

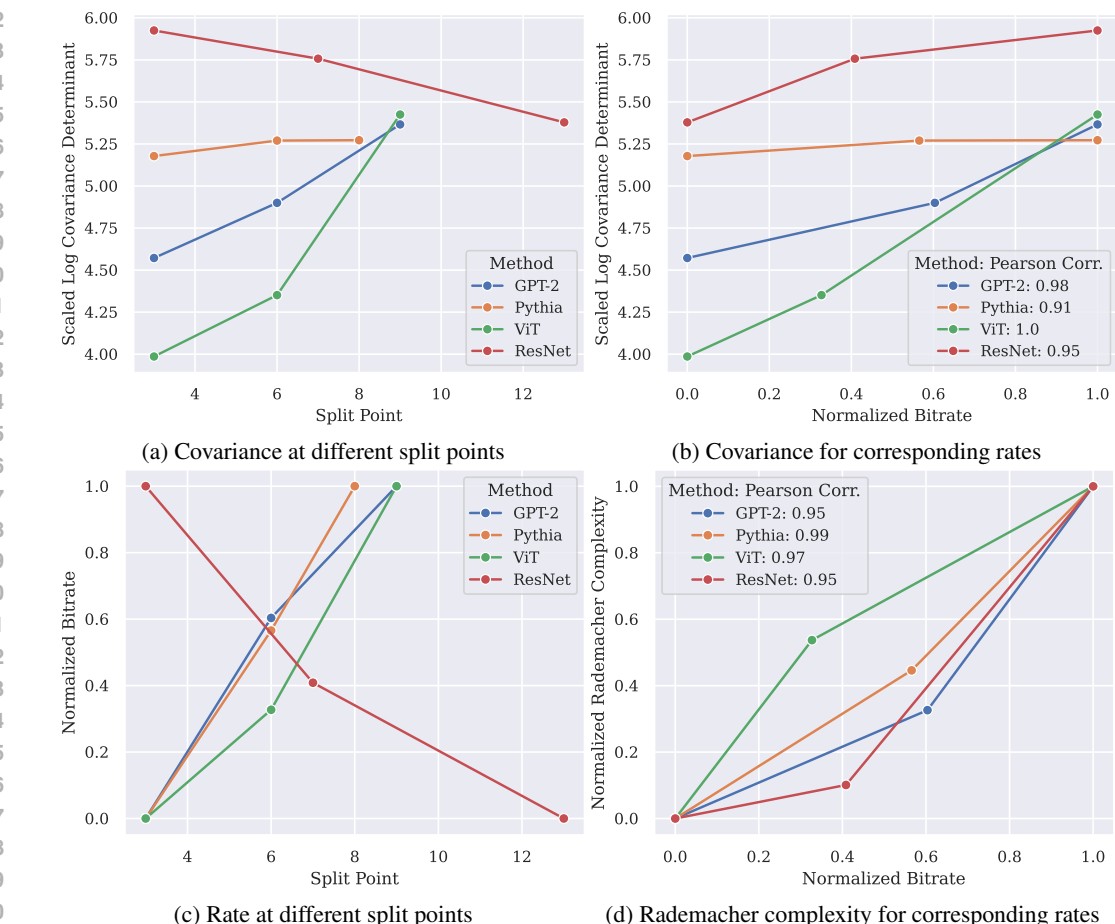

Figure 4: Rate, covariance determinant, and Rademacher complexity estimates at different split points, for GPT-2 Small, Pythia 160M, ViT B/16, and ResNet 34. Some axes are min-max scaled per method to facilitate comparison. The logarithmic scale is further scaled by $1/2D$. The Rademacher complexity and covariance determinant strongly correlate with rate. Only in ResNets, the rate-distortion performance increases with the split point.

## 4.4 RATE-DISTORTION PERFORMANCE AND THE SPLIT POINT

Figure 3b shows the rate-distortion performance of our proposed method for 3 equidistant split points $S$. A rate penalty on the representation at $S = 3$ results in better task performance than a model without a rate penalty. This behavior has been observed in the coding for machines literature, and it has been informally attributed to the reduction of irrelevant information to the task, resulting in simpler feature spaces that are easier to predict from.

The results also show an increase in rate-distortion as the split point is placed further from the source. This behavior could be unexpected since, the transformer being deterministic, the entropy of the processed input can only decrease. However, it is often observed in the context of coding for machines that rates operate at orders of magnitude higher than the theoretical bounds for the entropy of the target (Bajić, 2025). This behavior can be explained by an increase of the upper-bounds of the $\mathcal{V}$-entropy gap and its generalization error. As the source is further processed by non-linear functions in a high-dimensional space, the complexity of the intermediate representations increases, making their probability density more difficult to estimate.

To demonstrate this, we perform experiments in which, for different tasks and model architectures, we introduce a rate constraint at different split points, measure the rate obtained by the proposed entropy model, and compare it against estimates of the covariance determinant and the Rademacher complexity of the target representation. These two quantities are part of the $\mathcal{V}$-entropy bound in

Theorem 2 and the generalization bound in Theorem 3, respectively. The entropy model architecture, settings, and the optimization algorithm, are maintained the same across experiments, ensuring that the Lipschitz constant of the predictive family $\mathcal{V}_r$ *acts as a constant* in Theorem 3. We also estimate and compare the Lipschitz constants of the entropy models obtained, for further analysis.

We establish 4 methods for comparison: 1. The GPT-2 Small language models from Section 4.2; 2. Language models using the Pythia 160M architecture (Biderman et al., 2023); 3. A transformer-based image classification task using ViT B/16 (Dosovitskiy et al., 2021); and 4. A convolutional neural network image classification task using ResNet 34 (He et al., 2016). The goal behind these choices is to pinpoint which aspects (i.e., model architecture, modality) produce a decrease in rate-distortion performance in deeper information bottlenecks, and to evaluate the correlation between the rate and its bounds in these diverse scenarios.

See Appendix D for more details on the comparative methods, as well as information on how the estimates of covariance determinant and Rademacher complexity are computed. See Appendix D.4 for an analysis of the Lipschitz constants.

Figure 4a shows estimates of the covariance determinant of the target representation per split point, with strong correlations. Figure 4b shows the relationship between the covariance determinant of the target representation and the bitrate obtained by the proposed entropy model. The average Pearson correlation between these two measurements is 0.96. The strong correlation corroborates the increase in rate as the split point $S$ is placed further.

Figure 4c shows the change in rate as a function of the split point. For all transformer models, the rate increases with depth, whereas for the ResNet model, the inverse occurs. Figure 4d shows the estimate of the Rademacher complexity and compares it against the rate of the target representation. There is a strong Pearson correlation, with an average of 0.965.

## 5    SUMMARY AND CONCLUSION

The proposed entropy model outperforms more complex baselines by at least 10.7%. The empirical results also show that the task performance for early layers can be at most 17.8% better than rate-unconstrained models. We present a time analysis that justifies the value of rate-constrained representations for general lossless compression, and show there is an advantage in coding using the proposed entropy model when, under assumptions, the effective link speed is less than 37.77 Mbps.

The $\mathcal{V}$-entropy theory is extended by introducing the $\mathcal{V}$-entropy gap. For our architecture, we tied this concept to the rate term of our proposed loss. We presented bounds on the $\mathcal{V}$-entropy gap for diverse configurations of the side information, giving us insight into why this gap can increase. In addition, PAC-like bounds are presented for the generalization error of the $\mathcal{V}$-entropy gap. Together, these results extend the theory for rate-distortion optimization.

We see that for transformers using the proposed method, the rate-distortion performance decreases as we go into deeper layers. The behavior is justified by an increase in the Rademacher complexity and the covariance determinant of the target representation, concepts that bound the $\mathcal{V}$-entropy gap and its generalization error, respectively. A commensurate increase in the complexity of the entropy model to meet the requirements of the target representation could result in a decrease in the $\mathcal{V}$-entropy gap. However, this positive contribution is offset by an increase in the generalization error, which is upper-bound by the Lipschitz constant of the entropy model. An increase in this generalization error might explain why the more complex entropy models used as baselines in this work do not perform as well as our simpler proposed method.

We observe that only in the ResNet task considered, the covariance determinant and Rademacher complexity of the target representations decrease in deeper layers, along with their rate. We conclude that in transformers, the intermediate representations from deeper layers require an increase in complexity, which, as shown in this work, causes their rate to also increase. This transformer trait seems to be required in order to perform the tasks considered. We hope to develop potential solutions in future work.

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

# A  THEORETICAL RESULTS AND PROOFS

## A.1  THE $\mathcal{V}$-ENTROPY GAP IN TERMS OF KL DIVERGENCES

**Lemma 1.** *Let $\mathcal{V} \subseteq \Omega$ be a predictive family according to Xu et al. (2020), and $Y$ and $W$ two random variables with sample space $\mathcal{Y}$ and $\mathcal{W}$ respectively. Then:*

$$G_{\mathcal{V}}(Y|W) = \inf_{g \in \mathcal{V}} \mathbb{E}_{\mathbf{w} \sim W} \left[ \mathrm{KL} \left( P_{Y|\mathbf{w}} \| g[\mathbf{w}] \right) \right]. \tag{12}$$

*Proof.*

$$G_{\mathcal{V}}(Y|W) \triangleq |H_{\mathcal{V}}(Y|W) - H_{\Omega}(Y|W)| \tag{13}$$

$$= \left| \inf_{g \in \mathcal{V}} \mathbb{E}_{\mathbf{y},\mathbf{w} \sim Y,W} \left[ -\log g[\mathbf{w}](\mathbf{y}) \right] - \inf_{\omega \in \Omega} \mathbb{E}_{\mathbf{y},\mathbf{w} \sim Y,W} \left[ -\log \omega[\mathbf{w}](\mathbf{y}) \right] \right| \tag{14}$$

$$= \left| \inf_{g \in \mathcal{V}} \mathbb{E}_{\mathbf{y},\mathbf{w} \sim Y,W} \left[ -\log g[\mathbf{w}](\mathbf{y}) \right] - \mathbb{E}_{\mathbf{y},\mathbf{w} \sim Y,W} \left[ -\log P_{Y|W}(\mathbf{y}|\mathbf{w}) \right] \right| \tag{15}$$

$$= \inf_{g \in \mathcal{V}} \mathbb{E}_{\mathbf{y},\mathbf{w} \sim Y,W} \left[ -\log g[\mathbf{w}](\mathbf{y}) \right] - \mathbb{E}_{\mathbf{y},\mathbf{w} \sim Y,W} \left[ -\log P_{Y|W}(\mathbf{y}|\mathbf{w}) \right] \tag{16}$$

$$= \inf_{g \in \mathcal{V}} \mathbb{E}_{\mathbf{w} \sim W} \left[ \mathbb{E}_{\mathbf{y} \sim Y|\mathbf{w}} \left[ \log \frac{P_{Y|W}(\mathbf{y}|\mathbf{w})}{g[\mathbf{w}](\mathbf{y})} \right] \right] \tag{17}$$

$$= \inf_{g \in \mathcal{V}} \mathbb{E}_{\mathbf{w} \sim W} \left[ \mathrm{KL} \left( P_{Y|\mathbf{w}} \| g(\mathbf{w}) \right) \right]. \tag{18}$$

Eq. 15 was obtained due to $\forall \mathbf{w} \in \mathcal{W}, P_{Y|\mathbf{w}} \in \Omega$ and $H(P_{Y|\mathbf{w}}, P_{Y|\mathbf{w}}) \leq H(P_{Y|\mathbf{w}}, \omega[\mathbf{w}])$. $H(\cdot, \cdot)$ is the cross-entropy measure. Eq. 16 was obtained due to $\mathcal{V} \subseteq \Omega$. $\qquad\square$

## A.2  THE $\mathcal{V}$-ENTROPY GAP AS RATE

**Theorem 1.** *Let $\mathcal{V} \subseteq \Omega$ be a predictive family according to Xu et al. (2020), and let $Y = f(X)$, where $X$ is a random variable and $f$ is differentiable. Then:*

$$G_{\mathcal{V}}(Y|Y) = \inf_{g \in \mathcal{V}} \mathbb{E}_{\mathbf{x} \sim X} \left\{ -\log \left( g\left[ f(\mathbf{x}) \right] \circ f \right) (\mathbf{x}) \right\}. \tag{7}$$

*Proof.* Using Lemma 1, we have:

$$G_{\mathcal{V}}(Y|Y) = \inf_{g \in \mathcal{V}} \mathbb{E}_{\mathbf{y} \sim Y} \left[ \mathrm{KL} \left( P_{Y|\mathbf{y}} \| g[\mathbf{y}] \right) \right] \tag{19}$$

$$= \inf_{g \in \mathcal{V}} \mathbb{E}_{\mathbf{y} \sim Y} \left[ \mathbb{E}_{\mathbf{y} \sim Y|\mathbf{y}} \left[ \log \frac{P_{Y|\mathbf{y}}(\mathbf{y}|\mathbf{y})}{g[\mathbf{y}](\mathbf{y})} \right] \right] \tag{20}$$

$$= \inf_{g \in \mathcal{V}} \mathbb{E}_{\mathbf{y} \sim Y} \left[ -\log g[\mathbf{y}](\mathbf{y}) \right] \tag{21}$$

$$= \inf_{g \in \mathcal{V}} \mathbb{E}_{\mathbf{x} \sim X} \left\{ -\log \left( g\left[ f(\mathbf{x}) \right] \circ f \right) (\mathbf{x}) \right\}. \tag{22}$$

Eq. 21 uses $\forall \mathbf{y} \in \mathcal{Y}, P_{Y|\mathbf{y}}(\mathbf{y}|\mathbf{y}) = 1$. Eq. 22 changes variables considering that if $X$ is continuous and $Y$ is discrete, all probability density gets mapped to any of the symbols in $\mathcal{Y}$. $\qquad\square$

## A.3  COVARIANCE DETERMINANT BOUNDS FOR THE $\mathcal{V}$-ENTROPY GAP

**Theorem 4.** *Let $\mathcal{V} \subseteq \Omega$ be a predictive family according to Xu et al. (2020), and $Y$ a continuous random variable with covariance matrix $\Sigma$. Then, with $D = |Y| = T \times E$, we have:*

$$G_{\mathcal{V}}(Y|Y) \leq {}^1\!/_2 \log \det \Sigma + {}^D\!/_2 \log (2\pi e) + \inf_{g \in \mathcal{V}} \mathrm{KL} \left( P_Y \| g[\mathbf{y}] \right). \tag{23}$$

*Proof.* Using Lemma 1, we have:

$$G_{\mathcal{V}}(Y|Y) = \inf_{g \in \mathcal{V}} \mathbb{E}_{\mathbf{y} \sim Y} \left[ \text{KL} \left( P_{Y|\mathbf{y}} \| g[\mathbf{y}] \right) \right] \tag{24}$$

$$= \inf_{g \in \mathcal{V}} \mathbb{E}_{\mathbf{y} \sim Y} \left[ \mathbb{E}_{\mathbf{y} \sim Y|\mathbf{y}} \left[ \log \frac{P_{Y|\mathbf{y}}(\mathbf{y}|\mathbf{y})}{g[\mathbf{y}](\mathbf{y})} \right] \right] \tag{25}$$

$$= \inf_{g \in \mathcal{V}} \mathbb{E}_{\mathbf{y} \sim Y} \left[ \log \frac{1}{g[\mathbf{y}](\mathbf{y})} \right] \tag{26}$$

$$= \inf_{g \in \mathcal{V}} \mathbb{E}_{\mathbf{y} \sim Y} \left[ \log \frac{P_Y(\mathbf{y})}{g[\mathbf{y}](\mathbf{y}) P_Y(\mathbf{y})} \right] \tag{27}$$

$$= h(Y) + \inf_{g \in \mathcal{V}} \text{KL} \left( P_Y \| g[\mathbf{y}] \right) \tag{28}$$

$$\leq \frac{1}{2} \log \det \Sigma + \frac{D}{2} \log (2\pi e) + \inf_{g \in \mathcal{V}} \text{KL} \left( P_Y \| g[\mathbf{y}] \right). \tag{29}$$

Eq. 25 uses $\forall \mathbf{y} \in \mathcal{Y}, P_{Y|\mathbf{y}}(\mathbf{y}|\mathbf{y}) = 1$. Eq. 29 uses an upper bound on differential entropy (Cover & Thomas, 2006). $\square$

**Theorem 2.** *Let $\mathcal{V} \subseteq \Omega$ be a predictive family according to Xu et al. (2020), and $Y = q(Y'; \Delta)$ a quantized continuous random variable where $\text{cov}(Y') = \Sigma$, and $\Delta$ is the maximum quantization step. Then, with $D = |Y| = T \times E$, we have:*

$$G_{\mathcal{V}}(Y|Y) \leq \frac{1}{2} \log \det \Sigma + \frac{D}{2} \log (2\pi e) + \inf_{g \in \mathcal{V}} \text{KL} \left( P_Y \| g[\mathbf{y}] \right) - \log \Delta \quad as \; \Delta \to 0. \tag{9}$$

*Proof.* Using Lemma 1 and following the same first steps as in the proof for Theorem 4, we have:

$$G_{\mathcal{V}}(Y|Y) = H(Y) + \inf_{g \in \mathcal{V}} \text{KL} \left( P_Y \| g[\mathbf{y}] \right). \tag{30}$$

From this result, using $H(Y) \to h(Y') - \log \Delta$ as $\Delta \to 0$ (Cover & Thomas, 2006), we have:

$$\lim_{\Delta \to 0} G_{\mathcal{V}}(Y|Y) = h(Y') - \log \Delta + \inf_{g \in \mathcal{V}} \text{KL} \left( P_Y \| g[\mathbf{y}] \right) \tag{31}$$

$$\leq \frac{1}{2} \log \det \Sigma + \frac{D}{2} \log (2\pi e) + \inf_{g \in \mathcal{V}} \text{KL} \left( P_Y \| g[\mathbf{y}] \right) - \log \Delta. \tag{32}$$

Eq. 32 uses the same upper bound on differential entropy as Eq. 29. $\square$

### A.4 LIPSCHITZ BOUND FOR THE $\mathcal{V}$-ENTROPY GAP ASSUMING SIDE INFORMATION AS A FUNCTION OF A COMMON ANCESTOR

Before showing the main result of this section, we introduce the following lemma and its proof:

**Lemma 2.** *Given $Y = f(X), W = h(X)$, where $X$ is a continuous random variable, and $f, h$ are bijective and Lipschitz continuous with constants $L_f, L_h$ respectively, we have:*

$$\mathbb{E}_{\mathbf{x} \sim X} \left[ \log \frac{|J_{h(\mathbf{x})}|}{|J_{f(\mathbf{x})}|} \right] \leq D(L_h + L_{f^{-1}} - 2) \tag{33}$$

*Proof.*

$$\mathbb{E}_{\mathbf{x} \sim X} \left[ \log \frac{|J_{h(\mathbf{x})}|}{|J_{f(\mathbf{x})}|} \right] = \mathbb{E}_{\mathbf{x} \sim X} \left[ \log \frac{\left| \prod_{i=1}^{D} \lambda_i \left( J_{h(\mathbf{x})} \right) \right|}{\left| \prod_{i=1}^{D} \lambda_i \left( J_{f(\mathbf{x})} \right) \right|} \right] \tag{34}$$

$$= \mathbb{E}_{\mathbf{x} \sim X} \left[ \sum_{i=1}^{D} \log \left| \lambda_i \left( J_{h(\mathbf{x})} \right) \right| - \sum_{i=1}^{D} \log \left| \lambda_i \left( J_{f(\mathbf{x})} \right) \right| \right] \tag{35}$$

$$\leq \mathbb{E}_{\mathbf{x} \sim X} \left[ \sum_{i=1}^{D} \left\{ \left| \lambda_i \left( J_{h(\mathbf{x})} \right) \right| - 1 \right\} - \sum_{i=1}^{D} \left\{ 1 - \frac{1}{\left| \lambda_i \left( J_{f(\mathbf{x})} \right) \right|} \right\} \right] \tag{36}$$

$$= \mathbb{E}_{\mathbf{x} \sim X} \left[ \sum_{i=1}^{D} \left| \lambda_i \left( J_{h(\mathbf{x})} \right) \right| + \sum_{i=1}^{D} \frac{1}{\left| \lambda_i \left( J_{f(\mathbf{x})} \right) \right|} \right] - 2D \tag{37}$$

$$= \mathbb{E}_{\mathbf{x} \sim X} \left[ \left\| \boldsymbol{\lambda} \left( J_{h(\mathbf{x})} \right) \right\|_1 + \left\| \boldsymbol{\lambda} \left( J_{f(\mathbf{x})}^{-1} \right) \right\|_1 \right] - 2D \tag{38}$$

$$\leq \sqrt{D} \, \mathbb{E}_{\mathbf{x} \sim X} \left[ \left\| \boldsymbol{\lambda} \left( J_{h(\mathbf{x})} \right) \right\|_2 + \left\| \boldsymbol{\lambda} \left( J_{f(\mathbf{x})}^{-1} \right) \right\|_2 \right] - 2D \tag{39}$$

$$= \sqrt{D} \, \mathbb{E}_{\mathbf{x} \sim X} \left[ \left\| J_{h(\mathbf{x})} \right\|_{\mathrm{F}} + \left\| J_{f(\mathbf{x})}^{-1} \right\|_{\mathrm{F}} \right] - 2D \tag{40}$$

$$\leq D \, \mathbb{E}_{\mathbf{x} \sim X} \left[ \left\| J_{h(\mathbf{x})} \right\|_2 + \left\| J_{f(\mathbf{x})}^{-1} \right\|_2 \right] - 2D \tag{41}$$

$$= D \, \mathbb{E}_{\mathbf{x} \sim X} \left[ \left\| J_{h(\mathbf{x})} \right\|_2 + \left\| J_{f^{-1}(f(\mathbf{x}))} \right\|_2 \right] - 2D \tag{42}$$

$$\leq D(L_h + L_{f^{-1}} - 2). \tag{43}$$

Eq. 36 uses the logarithmic bounds $\log a \leq a - 1; \log a \geq 1 - 1/a$. Eq. 38 uses the fact that $\forall \lambda_i(A) \in \boldsymbol{\lambda}(A) \, \exists \, 1/\lambda_i(A) \in \boldsymbol{\lambda}(A^{-1})$. Eq. 39 uses the bound for the 1-norm $\|\mathbf{a}\|_1 \leq \sqrt{|\mathbf{a}|} \|\mathbf{a}\|_2$. Eq. 40 uses the bound for the 2-norm $\|\boldsymbol{\lambda}(A)\|_2 = \|A\|_{\mathrm{F}}$. Eq. 41 uses the assumption that $f$ is bijective and the Frobenius norm bound $\|A\|_{\mathrm{F}} \leq \sqrt{\mathrm{rank}(A)} \|A\|_2$. Eq. 42 uses the relationship between Jacobian inverses $J_{f(\mathbf{x})}^{-1} = J_{f^{-1}(f(\mathbf{x}))}$. Finally, Eq. 43 uses the Lipschitz continuity assumption of $f$ and $h$ and the Jacobian 2-norm bound $\left\| J_{f(\mathbf{x})} \right\|_2 \leq L_f \, \forall \mathbf{x} \in \mathbb{R}^D$. $\qquad\square$

Now we present the theorem:

**Theorem 5.** *Let $\mathcal{V} \subseteq \Omega$ be a predictive family according to Xu et al. (2020), $Y = f(X), W = h(X)$, where $X$ is a continuous random variable, and $f$, $h$ are bijective and Lipschitz continuous with constants $L_f$, $L_h$ respectively, and $\mathcal{Y}$, $\mathcal{W}$ are the sample spaces of $Y$, $W$ respectively. Then:*

$$G_{\mathcal{V}}(Y|W) \leq \inf_{g \in \mathcal{V}} \mathbb{E}_{\mathbf{y}, \mathbf{w} \sim Y, W} \left[ \log \frac{P_W(\mathbf{w})}{g[\mathbf{w}](\mathbf{y})} \right] + D(L_h + L_{f^{-1}} - 2), \tag{44}$$

*where $L_{f^{-1}}$ is the Lipschitz constant of the inverse of $f$.*

*Proof.* Since random variables $Y$ and $W$ share a common ancestor $X$, using the change of variables, their probabilities are related by:

$$P_Y(f(\mathbf{x})) = P_W(h(\mathbf{x})) \frac{|J_{h(\mathbf{x})}|}{|J_{f(\mathbf{x})}|}, \tag{45}$$

where $\left|J_{v(\mathbf{x})}\right|$ is the absolute Jacobian determinant of $v(\mathbf{x})$. Thus:

$$G_{\mathcal{V}}(Y|W) = \inf_{g \in \mathcal{V}} \mathbb{E}_{\mathbf{w} \sim W} \left[ \mathbb{E}_{\mathbf{y} \sim Y|\mathbf{w}} \left[ \log \frac{P_{Y|W}(\mathbf{y}|\mathbf{w})}{g[\mathbf{w}](\mathbf{y})} \right] \right] \tag{46}$$

$$= \inf_{g \in \mathcal{V}} \mathbb{E}_{\mathbf{w} \sim W} \left[ \mathbb{E}_{\mathbf{y} \sim Y|\mathbf{w}} \left[ \log \frac{P_{Y,W}(\mathbf{y}, \mathbf{w})}{g[\mathbf{w}](\mathbf{y})P_W(\mathbf{w})} \right] \right] \tag{47}$$

$$= \inf_{g \in \mathcal{V}} \mathbb{E}_{\mathbf{w} \sim W} \left[ \mathbb{E}_{\mathbf{y} \sim Y|\mathbf{w}} \left[ \log \frac{P_{W|Y}(\mathbf{w}|\mathbf{y})P_Y(\mathbf{y})}{g[\mathbf{w}](\mathbf{y})P_W(\mathbf{w})} \right] \right] \tag{48}$$

$$= \inf_{g \in \mathcal{V}} \mathbb{E}_{\mathbf{w},\mathbf{y} \sim W,Y} \left[ \log \frac{P_{W|Y}(\mathbf{w}|\mathbf{y})}{g[\mathbf{w}](\mathbf{y})} \right] + \mathbb{E}_{\mathbf{x} \sim X} \left[ \mathbb{E}_{\mathbf{y} \sim Y|h(\mathbf{x})} \left[ \log \frac{P_Y(\mathbf{y})}{P_W(h(\mathbf{x}))} \right] \right] \tag{49}$$

$$= \inf_{g \in \mathcal{V}} \mathbb{E}_{\mathbf{w},\mathbf{y} \sim W,Y} \left[ \log \frac{P_{W|Y}(\mathbf{w}|\mathbf{y})}{g[\mathbf{w}](\mathbf{y})} \right] + \mathbb{E}_{\mathbf{x} \sim X} \left[ \log \frac{\left|J_{h(\mathbf{x})}\right|}{\left|J_{f(\mathbf{x})}\right|} \right] \tag{50}$$

$$\leq \inf_{g \in \mathcal{V}} \mathbb{E}_{\mathbf{w},\mathbf{y} \sim W,Y} \left[ \log \frac{P_{W|Y}(\mathbf{w}|\mathbf{y})}{g[\mathbf{w}](\mathbf{y})} \right] + D(L_h + L_{f^{-1}} - 2) \tag{51}$$

$$\leq \inf_{g \in \mathcal{V}} \mathbb{E}_{\mathbf{w},\mathbf{y} \sim W,Y} \left[ \log \frac{P_W(\mathbf{w})}{g[\mathbf{w}](\mathbf{y})} \right] + D(L_h + L_{f^{-1}} - 2). \tag{52}$$

Eq. 50 uses Eq. 45. Eq. 51 uses Lemma 2. Eq. 52 uses $H(W|Y) \leq H(W)$. $\qquad\square$

The first term diminishes as the probability distributions of the side information and the rate of the target distribution agree. The second term cancels when $f$ and $h$ are identity functions.

### A.5 LIPSCHITZ BOUND FOR THE $\mathcal{V}$-ENTROPY GAP ASSUMING FIXED SIDE INFORMATION AS A FUNCTION OF THE TARGET REPRESENTATION

This formulation offers an alternative view in cases where the side information is considered fixed, such that the hyper-prior analysis transform $h$ is not part of the predictive family.

**Theorem 6.** *Let $\mathcal{V} \subseteq \Omega$ be a predictive family according to Xu et al. (2020), $Y = f(X), W = h(Y)$, where $X$ is a continuous random variable, $f$ and $h$ are bijective, $h$ is Lipschitz continuous with constant $L_h$, and $\mathcal{Y}, \mathcal{W}$ are the sample spaces of $Y, W$ respectively. Then:*

$$G_{\mathcal{V}}(Y|W) \leq \inf_{g \in \mathcal{V}} \mathbb{E}_{\mathbf{x} \sim X} \left[ \log \frac{P_W\left[(h \circ f)(\mathbf{x})\right]}{g\left[(h \circ f)(\mathbf{x})\right](f(\mathbf{x}))} \right] + D(L_h - 1). \tag{53}$$

*Proof.* Following the proof for Lemma 2 closely, we arrive at:

$$\mathbb{E}_{\mathbf{x} \sim X} \left[ \log \frac{\left|J_{(h \circ f)(\mathbf{x})}\right|}{\left|J_{f(\mathbf{x})}\right|} \right] = \mathbb{E}_{\mathbf{x} \sim X} \left[ \log \frac{\left|J_{h(f(\mathbf{x}))}J_{f(\mathbf{x})}\right|}{\left|J_{f(\mathbf{x})}\right|} \right] \tag{54}$$

$$= \mathbb{E}_{\mathbf{x} \sim X} \left[ \log \left|J_{h(f(\mathbf{x}))}\right| \right] \tag{55}$$

$$= \mathbb{E}_{\mathbf{x} \sim X} \left[ \log \left| \prod_{i=1}^{D} \lambda_i \left( J_{h(f(\mathbf{x}))} \right) \right| \right] \tag{56}$$

$$\leq \mathbb{E}_{\mathbf{x} \sim X} \left[ \sum_{i=1}^{D} \left| \lambda_i \left( J_{h(f(\mathbf{x}))} \right) \right| \right] - D \tag{57}$$

$$\leq D \, \mathbb{E}_{\mathbf{x} \sim X} \left[ \left\| J_{h(f(\mathbf{x}))} \right\|_2 \right] - D \tag{58}$$

$$\leq D(L_h - 1). \tag{59}$$

Plugging this result in Eq. 50 and changing variables with respect to $X$ arrives at the result. $\qquad\square$

The second term cancels when $h$ becomes the identity function. We could approximate $P_W(W)$ using the entropy model for the hyper-prior.

A.6  GENERALIZATION BOUNDS FOR THE $\mathcal{V}$-ENTROPY GAP

**Theorem 3.** *Let $\mathcal{V} \subseteq \Omega$ be a predictive family according to Xu et al. (2020), $Y$ a random variable with sample space $\mathcal{Y}$, and $\mathcal{D} = \{\mathbf{y}_i\}_{i=1}^N \sim Y$ a set of samples. Assume that $\forall g \in \mathcal{V}, \mathbf{y} \in \mathcal{Y}, \log g[\mathbf{y}](\mathbf{y}) \in [-B, B]$. Then, $\forall \delta \in (0, 1)$, with probability at least $1 - \delta$, we have:*

$$R_{\mathcal{V},\mathcal{D}}(Y|Y) \leq 2\operatorname{Lip}(\mathcal{V}_r)\operatorname{Rad}(\mathcal{D}) + B\sqrt{2/N \log 1/\delta}. \tag{11}$$

*where $\operatorname{Rad}$ is the Rademacher complexity of a set, and $\operatorname{Lip}(\mathcal{V}_r)$ is the maximum Lipschitz constant in $\mathcal{V}_r = \{v|v(\mathbf{y}) = \log g[\mathbf{y}](\mathbf{y}), g \in \mathcal{V}\}$.*

*Proof.* With $\hat{g} = \arg\min_{g \in \mathcal{V}} \sum_{\mathbf{y} \in \mathcal{D}} -\log g[\mathbf{y}](\mathbf{y})$, we derive:

$$R_{\mathcal{V},\mathcal{D}}(Y|Y) \triangleq \left| G_{\mathcal{V}}(Y|Y) - \inf_{g \in \mathcal{V}} \frac{1}{N} \sum_{\mathbf{y} \in \mathcal{D}} \log \frac{P_{Y|Y}(\mathbf{y}|\mathbf{y})}{g[\mathbf{y}](\mathbf{y})} \right| \tag{60}$$

$$= \left| H_{\mathcal{V}}(Y|Y) - H_{\Omega}(Y|Y) - \frac{1}{N} \sum_{\mathbf{y},\mathbf{w} \in \mathcal{D}} -\log \hat{g}[\mathbf{y}](\mathbf{y}) \right| \tag{61}$$

$$= \left| H_{\mathcal{V}}(Y|Y) - \frac{1}{N} \sum_{\mathbf{y} \in \mathcal{D}} -\log \hat{g}[\mathbf{y}](\mathbf{y}) \right| \tag{62}$$

$$\leq 2\operatorname{Rad}(\mathcal{V}_r \circ \mathcal{D}) + B\sqrt{2/N \log 1/\delta} \tag{63}$$

$$\leq 2\operatorname{Lip}(\mathcal{V}_r)\operatorname{Rad}(\mathcal{D}) + B\sqrt{2/N \log 1/\delta}. \tag{64}$$

Eq. 63 uses Lemma 3 in Xu et al. (2020). Eq. 64 uses the Kakade & Tewari Lemma (Kakade & Tewari, 2008) based on Talagrand's contraction principle (Ledoux & Talagrand, 2013; Bartlett & Mendelson, 2002). It states that if all vectors in a set $A$ are operated by a Lipschitz function, then $\operatorname{Rad}(A)$ is at most multiplied by the Lipschitz constant of the function. $\qquad\square$

**Theorem 7.** *Let $\mathcal{V} \subseteq \Omega$ be a predictive family according to Xu et al. (2020), $Y$ and $W$ random variables with sample spaces $\mathcal{Y}$ and $\mathcal{W}$, respectively, with $\mathcal{D} = \{(\mathbf{y}_i, \mathbf{w}_i)\}_{i=1}^N \sim Y, W$ as a set of samples. Assume that $\forall g \in \mathcal{V}, \mathbf{y} \in \mathcal{Y}, \mathbf{w} \in \mathcal{W}, \log g[\mathbf{w}](\mathbf{y}) \in [-B, B], \log P_{Y|W}(\mathbf{y}|\mathbf{w}) \in [-B, B]$, and that the Lipschitz constant of $P_{Y|\mathbf{w}} \forall \mathbf{w} \in \mathcal{W}$ is governed by the Lipschitz constant of a multivariate normal distribution with covariance matrix $\alpha I$, such that $\alpha \leq \exp(2BD^{-1})(2\pi)^{-1}$. Then, $\forall \delta \in (0, 1)$, with probability at least $1 - \delta$, we have:*

$$R_{\mathcal{V},\mathcal{D}}(Y|W) \leq 2\left[(\operatorname{Lip}(\mathcal{V}_r) + \beta)\operatorname{Rad}(\mathcal{D}) + B\sqrt{2/N \log 1/\delta}\right], \tag{65}$$

*where $\operatorname{Rad}(\mathcal{D})$ is the Rademacher complexity of the concatenated samples in the dataset $\mathcal{D}$, $\operatorname{Lip}(\mathcal{V}_r)$ is the maximum Lipschitz constant in $\mathcal{V}_r = \{v|v(\mathbf{w}, \mathbf{y}) = \log g[\mathbf{w}](\mathbf{y}), g \in \mathcal{V}\}$, and:*

$$\beta \triangleq \alpha^{-3/2}\left[2B - D\log(2\pi\alpha)\right]^{1/2}. \tag{66}$$

*Proof.* The assumption that $\log P_{Y|w} \in [-B, B]$ implies that:

$$-1/2(\mathbf{y} - \boldsymbol{\mu})^\top (\alpha I)^{-1}(\mathbf{y} - \boldsymbol{\mu}) - D/2\log(2\pi\alpha) \geq -B \tag{67}$$

$$\implies \|\mathbf{y} - \boldsymbol{\mu}\|_2 \leq \alpha^{-1/2}\left[2B - D\log(2\pi\alpha)\right]^{1/2}, \tag{68}$$

where, to keep the right-hand-side positive as required by the norm, implies:

$$2B - D\log(2\pi\alpha) \geq 0 \implies \alpha \leq \exp(2BD^{-1})(2\pi)^{-1}. \tag{69}$$

The Lipschitz constant of $\log P_{Y|w}$ under the assumed constraint is upper bounded as:

$$\text{Lip}(\log P_{Y|w}) = \sup_{\mathbf{y} \in \mathbb{R}^D} \|\nabla \log \mathcal{N}(\mathbf{y}; \boldsymbol{\mu}, \alpha I)\|_\infty \tag{70}$$

$$= \alpha^{-1} \sup_{\mathbf{y} \in \mathbb{R}^D} \|\mathbf{y} - \boldsymbol{\mu}\|_\infty \tag{71}$$

$$\leq \alpha^{-1} \sup_{\mathbf{y} \in \mathbb{R}^D} \|\mathbf{y} - \boldsymbol{\mu}\|_2 \tag{72}$$

$$\leq \alpha^{-3/2} \left[ 2B - D \log(2\pi\alpha) \right]^{1/2} \tag{73}$$

$$\triangleq \beta. \tag{74}$$

Eq. 73 is obtained by plugging in Eq. 68. Finally, with $\hat{g} = \arg\min_{g \in \mathcal{V}} \sum_{(\mathbf{y}, \mathbf{w}) \in \mathcal{D}} -\log g[\mathbf{w}](\mathbf{y})$, we derive:

$$R_{\mathcal{V}, \mathcal{D}}(Y|W) \tag{75}$$

$$\triangleq \left| G_\mathcal{V}(Y|W) - \inf_{g \in \mathcal{V}} \frac{1}{N} \sum_{(\mathbf{y}, \mathbf{w}) \in \mathcal{D}} \log \frac{P_{Y|W}(\mathbf{y}|\mathbf{w})}{g[\mathbf{w}](\mathbf{y})} \right| \tag{76}$$

$$= \left| H_\mathcal{V}(Y|W) - H_\Omega(Y|W) - \frac{1}{N} \sum_{(\mathbf{y}, \mathbf{w}) \in \mathcal{D}} \log \frac{P_{Y|W}(\mathbf{y}|\mathbf{w})}{\hat{g}[\mathbf{w}](\mathbf{y})} \right| \tag{77}$$

$$\leq \left| H_\mathcal{V}(Y|W) - \frac{1}{N} \sum_{(\mathbf{y}, \mathbf{w}) \in \mathcal{D}} -\log \hat{g}[\mathbf{w}](\mathbf{y}) \right| + \left| \frac{1}{N} \sum_{(\mathbf{y}, \mathbf{w}) \in \mathcal{D}} -\log P_{Y|W}(\mathbf{y}|\mathbf{w}) - H_\Omega(Y|W) \right| \tag{78}$$

$$\leq 2\,\text{Rad}(\mathcal{V}_r \circ \mathcal{D}) + 2\,\text{Rad}(\log P_{Y|W} \circ \mathcal{D}) + 2B\sqrt{2/N \log 1/\delta} \tag{79}$$

$$\leq 2\,\text{Lip}(\mathcal{V}_r)\,\text{Rad}(\mathcal{D}) + 2\,\text{Lip}(\log P_{Y|W})\,\text{Rad}(\mathcal{D}) + 2B\sqrt{2/N \log 1/\delta} \tag{80}$$

$$\leq 2\left[ \text{Lip}(\mathcal{V}_r)\,\text{Rad}(\mathcal{D}) + \beta\,\text{Rad}(\mathcal{D}) + B\sqrt{2/N \log 1/\delta} \right] \tag{81}$$

$$= 2\left[ (\text{Lip}(\mathcal{V}_r) + \beta)\,\text{Rad}(\mathcal{D}) + B\sqrt{2/N \log 1/\delta} \right]. \tag{82}$$

Eq. 78 uses the triangle inequality. Eq. 79 uses Lemma 3 in Xu et al. (2020) on each of the two terms, where the predictive family for the second term has been reduced to $\{P_{Y|W}\}$. Eq. 80 uses the Kakade & Tewari Lemma previously introduced. Eq. 81 uses Eq. 74. Similarly to $\text{Lip}(\mathcal{V}_r)$, $\text{Lip}(\log P_{Y|W})$ corresponds to the largest Lipschitz constant in $\{\log P_{Y|w}, w \in \mathcal{W}\}$. $\qquad \square$

### A.7 MATHEMATICAL NOTATION

Table 1 compiles the most relevant mathematical notation used in this work.

## B RATE-DISTORTION PERFORMANCE BENCHMARKS: ADDITIONAL RESULTS AND DETAILS

### B.1 FOURIER BASIS DENSITY MODEL FOR THE HYPER-PRIOR

As a benchmark method, the assumed PDF for each dimension of the hyper-prior is independently modeled as a Fourier series with a finite number of coefficients. To ensure non-negativity of the PDF, the coefficients are auto-correlated, making the Fourier series positive semi-definite. The function over one period is divided by its integral for normalization, and has a closed-form solution. The resulting periodic density function is extended to the entire real line by a learnable mapping $(-1, 1) \to \mathbb{R}$ parameterized by a scaling and an offset learnable parameter.

The probability of a symbol is computed following Eq. 1, where $g_w$ has been replaced by this density model. The coefficients are learned while training the proposed loss function. We use 60 coefficients for a total of 120 parameters when counting the real and imaginary components. This number is similar to the amount of parameters used in the entropy model of the hyper-prior in the proposed method. The hyper-prior $W$ is quantized using the same approach in the proposed architecture.

Table 1: Notation reference

| Notation | Definition |
| --- | --- |
| $\mathbf{a} \sim A$ | Radamacher random variable with sample space $\{-1, 1\}^N$ |
| $\mathcal{A}$ | $M$ samples of $A$: $\mathcal{A} = \{\mathbf{a}_i\}_{i=1}^M \sim A$ |
| $B$ | Bound such that $\log g[\mathbf{w}](\mathbf{y}), \log P_{Y\|W}(\mathbf{y}\|\mathbf{w}) \in [-B, B]$ |
| $C$ | Embedding size of the side information: $W \in \mathbb{R}^{T \times C}$ |
| $D$ | Number of elements in $Y$ such that $D = \|Y\| = T \times E$ |
| $\mathcal{D}$ | $N$ data samples: $\mathcal{D} = \{\mathbf{y}_i\}_{i=1}^N \sim Y$ |
| $d(\hat{\mathbf{z}}, \mathbf{z})$ | Task loss function or distortion function |
| $E$ | Embedding size of the target representation: $Y \in \mathbb{R}^{T \times E}$ |
| $f_{1,S}(X)$ | Function producing the target representation $Y$ |
| $f_{S+1,L}(Y)$ | Second split/module of the neural network, without header layers |
| $f_l(\cdot)$ | Transformer block: $\{f_l : \mathbb{R}^{T \times E} \to \mathbb{R}^{T \times E}\}_{l=1}^L$ |
| $G_\mathcal{V}(\cdot\|\cdot)$ | The $\mathcal{V}$-entropy gap, Eq. 6 |
| $g_\mu(W)$ | Means produced by $g_y$, $g_\mu : \mathcal{W} \to \mathbb{R}^{T \times E}$ |
| $g_\sigma(W)$ | Variances produced by $g_y$, $g_\sigma : \mathcal{W} \to \mathbb{R}_+^{T \times E}$ |
| $g_y(W)$ | Entropy model for the target representation $g_y : \mathcal{W} \to \mathcal{N}(\mathcal{Y})$ |
| $g_w(w_{i,j}; \boldsymbol{\theta}_j)$ | Entropy model for the hyper-prior $g_w : \mathbb{R} \to [0, 1]$ |
| $H_\mathcal{V}(\cdot\|\cdot)$ | Conditional $\mathcal{V}$-entropy, Eq. 2 |
| $H(\cdot)$ | Shannon's entropy |
| $H(\cdot, \cdot)$ | Cross-entropy |
| $h(Y)$ | Hyper-prior model $h : \mathcal{Y} \to \mathbb{R}^{T \times C}$ |
| $I$ | Identity matrix |
| $J_r(\mathbf{y})$ | Jacobian matrix of function $v$ evaluated at $\mathbf{y}$ |
| $\mathrm{KL}(\cdot\|\|\cdot)$ | Kullback–Leibler divergence |
| $L$ | Number of transformer blocks in a transformer-based neural network |
| $\mathcal{L}(Y, W, \hat{Z}, Z)$ | Proposed loss function, Eq. 5 |
| $M$ | Number of samples from $A$: $\|\mathcal{A}\|$ |
| $\mathcal{N}, \mathcal{N}(\mathcal{Y})$ | Normal PDF, or the set of all normal PDFs on $\mathcal{Y}$ |
| $N$ | Dataset size $\|\mathcal{D}\|$ |
| $q(\cdot)$ | Quantization function |
| $R_{\mathcal{V},\mathcal{D}}(\cdot\|\cdot)$ | Generalization error of the $\mathcal{V}$-entropy gap, Eq. 10 |
| $r_y(\mathbf{y}; \mathbf{w})$ | Rate function for the target representation, Eq. 3 |
| $r_w(\mathbf{w})$ | Rate function for the side-information, Eq. 1 |
| $S$ | Split point $S \in \{1, ..., L\}$ |
| $T$ | Target representation context size: $Y \in \mathbb{R}^{T \times E}$ |
| $t(\cdot)$ | $t : \mathbb{R}^{T \times E} \to \mathcal{Z}$, head module producing predictions $\hat{Z}$ for target $Z$ |
| $\mathcal{V}$ | Predictive family (Xu et al., 2020) |
| $\mathcal{V}_r$ | Set of log probability functions of $\mathcal{V}$: $\{v\|v(\mathbf{w}, \mathbf{y}) = \log g[\mathbf{w}](\mathbf{y}), g \in \mathcal{V}\}$ |
| $\mathbf{w} \sim W, \mathbf{w} \in \mathcal{W}$ | Side information with sample space $\mathcal{W} \subseteq \mathbb{R}^{T \times C}$ |
| $\mathbf{x} \sim X, \mathbf{x} \in \mathcal{X}$ | Input with $\mathcal{X} \subseteq \mathbb{R}^{T \times E}$ |
| $\mathbf{y} \sim Y, \mathbf{y} \in \mathcal{Y}$ | Target representation with $\mathcal{Y} \subseteq \mathbb{R}^{T \times C}$ |
| $\mathbf{z} \sim Z, \mathbf{z} \in \mathcal{Z}$ | Task target |
| $\hat{\mathbf{z}} \sim \hat{Z}, \hat{\mathbf{z}} \in \mathcal{Z}$ | Model prediction |
| $\alpha$ | Variance factor in Theorem 7, where $\alpha \leq \exp(2BD^{-1})(2\pi)^{-1}$ |
| $\Phi$ | Normal CDF |
| $\Lambda$ | The set of all diagonal covariance matrices |
| $\lambda$ | Rate-distortion trade-off parameter |
| $\Theta$ | Parameters of the hyper-prior entropy model: $\Theta = \{\boldsymbol{\theta}_j, ..., \boldsymbol{\theta}_C\}$ |
| $\Omega$ | Set of all probability functions over $\mathcal{Y}$ such that $\Omega = \{\mathcal{W} \cup \{\oslash\} \to \mathcal{P}(\mathcal{Y})\}$ |
| $\Delta$ | Quantization step |

## B.2 DIRECT-ACCESS ENTROPY MODEL

In another benchmark method, an auto-regressive transformer predicts the means and variances of the target representation using the hyper-prior *and the previous elements in the time series* of the

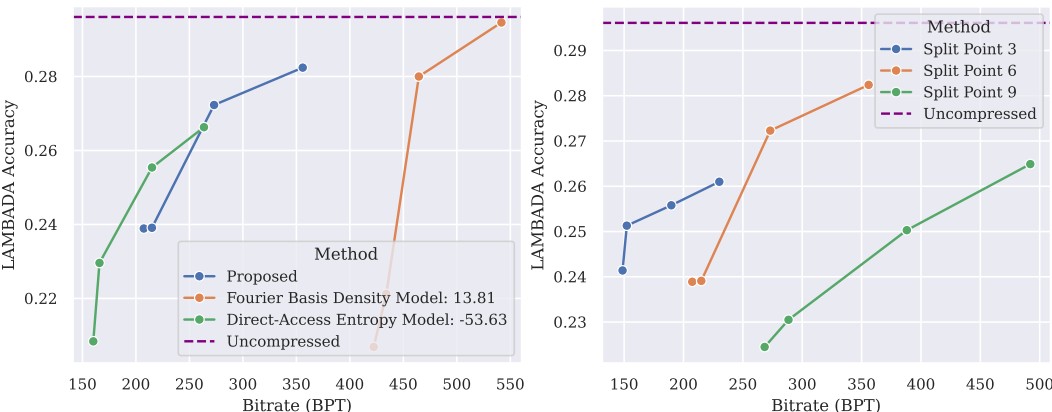

(a) Split point 6 under different entropy models      (b) Split points 3, 6, and 9 using the proposed method

Figure 5: Rate-performance for GPT-2 evaluated on the the LAMBADA language task. The rate is measured in bits-per-token (BPT). *Uncompressed* is a model with no quantization or rate penalty ($\lambda = 0$).

target representation. The hyper-prior $W$ is used as query embeddings and additional dimensions for the key and value embeddings of the attention mechanism in an initial transformer block. The auto-regressive restriction allows the entropy model to be used for coding since the serialized decoding process only has access to previously-decoded elements.

Figure 2b presents an overview of the architecture. Using the same proposed hyper-prior model and corresponding entropy model, the entropy model for the target representation first passes the hyper-prior through 4 transformer blocks with causal attention masks that maintain the same embedding size $C$. The resulting tensor is then processed by a custom transformer block in which, for each role, a single dense layer produces: 1. Query embeddings of $C$ dimensions; 2. Key and value embeddings of size $C$ and $E$ respectively, as a function of the target embeddings and this tensor; and 3. The value embedding of the first time step after attention. This transformer block uses an attention mask that prevents access to elements from the current time step forward. The output of the transformer block is then further processed by 3 more transformer blocks that use causal attention masks and retain the same embedding size $E$, except for the last block, which increases the output embedding size to $2E$. The output tensor is split into the means and variances of the multivariate normal distribution for $Y$.

### B.3 TASK PERFORMANCE

A more interpretable task metric for a language model is LAMBADA (Paperno et al., 2016), which evaluates the ability to comprehend context and understand discourse. We score the language models created in Section 4.2 using this method. Since the language models have not been trained further for question-answering tasks, we strip all prompting, making no distinction between context and target sentences, and use the first word produced after stripping punctuation.

Figure 5a shows the rate-performance of the proposed entropy model against the baselines introduced in this work. There is some missing overlap between the curves, making the BD-rate measurements unreliable. Even so, there is a disadvantage for the direct-access entropy model against the proposed entropy model of -53.63%. The Fourier basis density model has an advantage of 13.81% over the proposed method.

Figure 5b shows the rate-performance of the proposed entropy model at different split points. Interestingly, some points for the split point $S = 6$ have the best overall LAMBADA accuracies.

Tables 2, 3, and 4 provide the task performances achieved for the models and the corresponding tasks evaluated in Section 4.

Table 2: GPT-2 rate-distortion performance

| Model | Split Pt. | $\lambda$ | Hyper-Prior BPT | Total BPT | Perplexity | LAMBADA |
|---|---|---|---|---|---|---|
| Proposed | 3 | 0.001 | 41.43 | 230.16 | 16.98 | 0.2610 |
| Proposed | 3 | 0.025 | 39.20 | 189.74 | 17.42 | 0.2558 |
| Proposed | 3 | 0.075 | 33.63 | 152.25 | 18.36 | 0.2513 |
| Proposed | 3 | 0.010 | 33.97 | 148.68 | 19.10 | 0.2414 |
| Proposed | 6 | 0.001 | 45.55 | 355.93 | 20.89 | 0.2824 |
| Proposed | 6 | 0.025 | 40.48 | 273.06 | 21.25 | 0.2723 |
| Proposed | 6 | 0.075 | 36.31 | 214.98 | 22.69 | 0.2391 |
| Proposed | 6 | 0.010 | 35.06 | 207.36 | 22.88 | 0.2389 |
| Proposed | 9 | 0.001 | 52.71 | 492.08 | 21.61 | 0.2649 |
| Proposed | 9 | 0.025 | 47.01 | 388.11 | 22.80 | 0.2503 |
| Proposed | 9 | 0.075 | 40.53 | 288.44 | 24.59 | 0.2305 |
| Proposed | 9 | 0.010 | 39.03 | 268.34 | 25.12 | 0.2245 |
| Fourier | 6 | 0.001 | 249.18 | 541.67 | 21.15 | 0.2946 |
| Fourier | 6 | 0.025 | 248.14 | 464.49 | 21.70 | 0.2800 |
| Fourier | 6 | 0.075 | 309.20 | 434.07 | 24.03 | 0.2212 |
| Fourier | 6 | 0.010 | 302.62 | 422.32 | 24.19 | 0.2069 |
| Direct-Access | 6 | 0.001 | 32.89 | 263.62 | 22.43 | 0.2663 |
| Direct-Access | 6 | 0.025 | 29.91 | 215.04 | 22.91 | 0.2554 |
| Direct-Access | 6 | 0.075 | 30.47 | 165.94 | 25.22 | 0.2296 |
| Direct-Access | 6 | 0.010 | 30.24 | 160.16 | 26.24 | 0.2084 |

Table 3: Split point rate-distortion performance for language models

| Model | Split Pt. | $\lambda$ | BPT | Perplexity | $\mathrm{Rad}_{\mathcal{D}}(f)$ | $\log \mathrm{Lip}(v)$ | $1/2D \log |\mathrm{Cov}(Y)|$ |
|---|---|---|---|---|---|---|---|
| GPT-2 | 3 | 0.010 | 148.7 | 19.10 | 1.18 | 6.66 | 4.57 |
| GPT-2 | 6 | 0.001 | 355.9 | 20.89 | 5.26 | 4.19 | 4.90 |
| GPT-2 | 9 | 0.001 | 492.1 | 21.61 | 13.69 | 5.18 | 5.35 |
| Pythia | 3 | 0.001 | 371.5 | 21.36 | 10.12 | 5.05 | 5.16 |
| Pythia | 6 | 0.001 | 451.9 | 21.99 | 12.05 | 5.85 | 5.27 |
| Pythia | 9 | 0.001 | 557.1 | 22.39 | 14.46 | 4.63 | 5.35 |

Table 4: Split point rate-distortion performance for image classification

| Model | Split Pt. | $\lambda$ | BPP | Accuracy | $\mathrm{Rad}_{\mathcal{D}}(f)$ | $\log \mathrm{Lip}(v)$ | $1/2D \log |\mathrm{Cov}(Y)|$ |
|---|---|---|---|---|---|---|---|
| ViT | 3 | 0.01 | 2.35 | 0.79 | 0.12 | 4.20 | 3.99 |
| ViT | 6 | 0.01 | 3.27 | 0.79 | 0.31 | 4.15 | 4.35 |
| ViT | 9 | 0.01 | 5.15 | 0.79 | 0.47 | 3.53 | 5.42 |
| ResNet | 3 | 0.01 | 4.75 | 0.68 | 1.88 | 4.13 | 5.93 |
| ResNet | 7 | 0.01 | 2.65 | 0.67 | 1.13 | 4.13 | 5.76 |
| ResNet | 13 | 0.01 | 1.20 | 0.67 | 1.04 | 3.99 | 5.38 |

## B.4 EXPERIMENTAL SETTINGS AND RESOURCE DETAILS

The text is processed using the TikToken GPT-2 tokenizer (Solaiman et al., 2019) and each sample has $T = 1024$ tokens, where different documents in the sequence are separated by a special token. Following (Karpathy, 2022), we use a linear warmup followed by a cosine learning rate schedule for AdamW (Loshchilov & Hutter, 2019), with coefficients $\beta_1 = 0.9, \beta_2 = 0.95$, and a weight decay of 0.1 placed on the two-dimensional parameter tensors of the language model. Using gradient accumulation, each optimization step uses 480 samples.

For convenience, all rate values reported correspond to estimates from the model. We noticed no change in distortion and less than 1% increase in rate when using the arithmetic coder, which could mitigated with a small sacrifice in speed.

All models were trained on a single NVIDIA A40 GPU. During training, models required at most 20 GB of VRAM. All models were trained with bfloat16 precision except for the ones using the ResNet architecture. The estimates of Rademacher complexities, Lipschitz constants, and covariance determinants were computed using a single NVIDIA A40 GPU, on less than 5 GB of VRAM, taking at most 1 hour per measure and model. An estimate of 320 GPU hours were spent on preliminary experiments.

We use the OpenWebText split provided in (Gokaslan et al., 2019). The dataset is provided under the CC0 license. All results pertaining this dataset are reported on the validation set. The OpenWebText is treated as a contiguous text file and a sample is a random window of text with a context size of 1,024. An epoch is considered to be 1,000 gradient descent steps of 480,000 random samples. The validation set consists of 48,000 random samples.

The ImageNet-1k (Russakovsky et al., 2015) is under a custom non-commercial license. Tasks trained on the dataset use the original splits. Each epoch trains on a subset of 100,000 random samples from the training set. All pertaining results are reported on the validation set. Random augmentations such as shearing, translation, rotation, and color jittering are applied to the samples, and MixUp and CutMix transformations are applied to a batch of size 16.

To produce the rate-distortion curves for the GPT-2 language models, first, a model is trained from scratch with $\lambda = 0.0001$ for 50 epochs, or until no improvement has been attained for more than 5 epochs. This takes around 120 hours. Finally, training is restarted for each $\lambda \in \{0.001, 0.0025, 0.0075, 0.01\}$ from the weights initially obtained. This training is done for around 25 epochs, or until no improvement has been attained for more than 5 epochs. This process usually takes 48 hours. For higher values of $\lambda$, the loss can diverge during training. In such cases, the maximum learning rate is set to 0.0001 and training is restarted from the checkpoint achieving the lowest loss.

For the Pythia language models, we use $\lambda = 0.001$. ViT and ResNet use $\lambda = 0.01$ on all experiments. The rate used as a loss term for the language models is computed in terms of bits-per-token (BPT), whereas the image classifiers use bits-per-pixel (BPP). The training of the Pythia languages models is started from the weights provided in (Biderman et al., 2023), under the Apache License, version 2.0. The ViT and ResNet uses the weights provided in (TorchVision maintainers and contributors, 2016), under the BSD 3-Clause license.

Additional hyper-parameters are reported on Table 5.

## C  BITRATE AND SPEED COMPARISONS: SETTINGS AND ADDITIONAL DETAILS

Deflate uses the LZ77 compression algorithm and Huffman coding. It is used in the ZIP and PNG file formats. We compare results on the same 1,000 validation samples from OpenWebText. We use the Python built-in version of this codec, which is implemented using bindings to the *zlib* C library. Our implementation of the proposed method is not optimized. The hyper-prior and entropy model ran on an NVIDIA 2080 RTX Ti GPU. The arithmetic coder ran on one core of an Intel Core i9-9900K CPU. The Deflate and Zstandard algorithms ran on the same system using one CPU core.

## D  RATE-DISTORTION PERFORMANCE AND THE SPLIT POINT: ADDITIONAL DETAILS AND RESULTS

### D.1  COMPARATIVE METHODS

The language modeling tasks use the same dataset and settings as discussed in Section 4.2. The image classification tasks use ImageNet-1k (Russakovsky et al., 2015) and the corresponding settings are similar to those well-established in TorchVision (TorchVision maintainers and contributors, 2016), including data augmentation, loss functions, and training methods, their parameters and

Table 5: Hyper-parameter settings

| Parameter | GPT-2 | Pythia | ViT | ResNet |
|---|---|---|---|---|
| Precision | bfloat16 | bfloat16 | bfloat16 | float32 |
| Target representation dimensionality ($E$) | 768 | 768 | 768 | 768 |
| Side information dimensionality ($C$) | 24 | 24 | 24 | 24 |
| Target representation context size ($T$) | 1,024 | 1,024 | 49 | 49 |
| Side information density parameters ($|\boldsymbol{\theta}_j|$) | 118 | 118 | 118 | 118 |
| Distortion function | cross-entr. | cross-entr. | cross-entr. | cross-entr. |
| Tokenizer | GPT-2 | GPT-2 | – | – |
| Label smoothing | – | – | 0.11 | 0 |
| MixUp coefficient | – | – | 0.2 | 0.2 |
| CutMix coefficient | – | – | 1 | 1 |
| Random augmentations | – | – | 2 | 2 |
| Augmentation magnitude | – | – | 9 | 9 |
| Batch size | 12 | 12 | 16 | 16 |
| Accumulated gradient batches | 40 | 40 | 40 | 40 |
| Optimizer | AdamW | AdamW | AdamW | AdamW |
| Optimizer parameters | (0.9, 0.95) | (0.9, 0.95) | (0.9, 0.95) | (0.9, 0.95) |
| Maximum learning rate | 0.0006 | 0.0006 | 0.0006 | 0.0006 |
| Minimum learning rate | 0.00006 | 0.00006 | 0.00006 | 0.00006 |
| Weight decay | 0.1 | 0.1 | 0.3 | 0.001 |
| Patience | 5 | 5 | 5 | 5 |
| Gradient norm clipping | 1 | 1 | 1 | 1 |
| Warm up function | linear | linear | linear | linear |
| Warm up steps | 2,000 | 2,000 | 2,000 | 2,000 |
| Schedule function | cosine | cosine | cosine | cosine |
| Maximum steps | 600,000 | 600,000 | 600,000 | 600,000 |

schedules. For methods 1-3, split points 3, 6, and 9 are evaluated. For the ResNet method, we evaluate split points at blocks 3, 7, and 13.

To adapt our entropy model to a ResNet, a convolutional layer is prepended to the entropy model so, across all split points, the resulting number of channels is $E = 768$ and the spatial dimensions are $7 \times 7$. These dimensions are flattened out and used as input for the entropy model after learnable positional embeddings have been added. A dense layer is appended to the proposed entropy model to recover the initial dimensionality.

The $\lambda$ hyper-parameter is chosen so that the distortion obtained at subsequent split points is as close and higher than previous points. Thus, the changes in rate cannot be attributed to different distortions.

## D.2 COVARIANCE DETERMINANT OF THE TARGET REPRESENTATION

We compute an approximation of the covariance determinant of the target representation $Y$ using the eigenvalues of the Hessenberg matrix produced by 1,000 Arnoldi iterations (Stewart, 2002) over $N = 1,000$ samples, with context size $T = 512$. The Arnoldi iteration algorithm produces a Krylov subspace for a matrix and a Hessenberg matrix with the dot products of the vectors in this subspace. It is often observed that the eigenvalues of this Hessenberg matrix converge to eigenvalues of the original matrix. Typically, its largest eigenvalues. We use these eigenvalues as an estimation of the covariance determinant of the target representation.

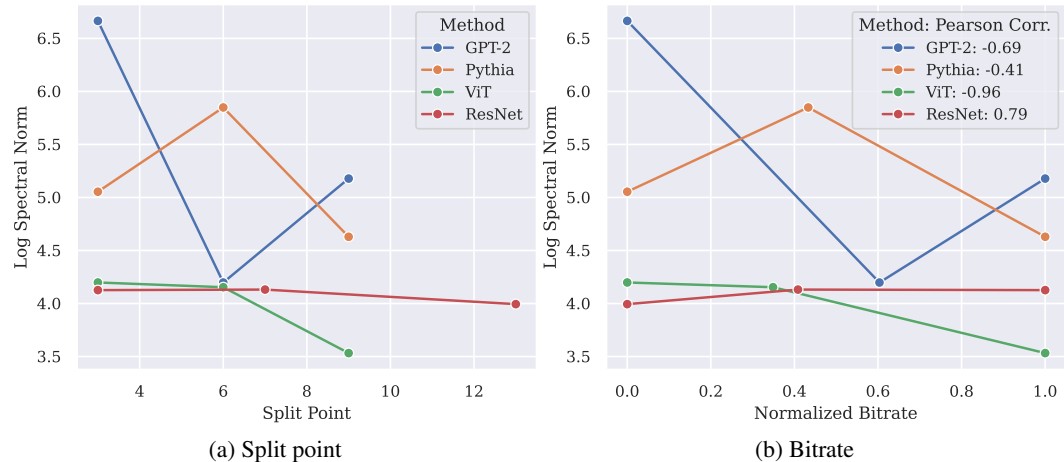

Figure 6: Estimates of the Lipschitz constant at different split points and corresponding bitrates, for GPT-2 Small, Pythia 160M, ViT B/16 and ResNet 34. The logarithmic scale is used.

### D.3 RADEMACHER COMPLEXITY OF THE TARGET REPRESENTATION

To estimate the Rademacher complexity of the target representation, we replace the Rademacher random variable expectation in its definition with an empirical measure, obtaining:

$$\bar{\mathrm{Rad}}(\mathcal{D}) = \frac{1}{MN} \sum_{\mathbf{a} \in \mathcal{A}} \max_{i=1}^{T} \max_{j=1}^{E} \left( \left| \sum_{k=1}^{N} a_k \mathcal{D}_k \right| \right)_{i,j}, \tag{83}$$

where $\mathcal{D}_k$ indexes the samples of $Y$, and $\mathcal{A} = \{\mathbf{a}_i\}_{i=1}^{M} \sim A$, where $A$ is a random variable with sample space $\{-1, 1\}^N$ following the Rademacher distribution. We set $N = 1,000, M = 10,000$ and the context size to $T = 512$. Since our estimate of Rademacher complexity reacts to changes in dimensionality, the measure for the ResNet method is performed on the output of the convolution layer prepended to the entropy model, which has the same dimensionality across split points.

The target representation must be quantized before computing the estimates of Rademacher complexity and covariance determinant. The Rademacher complexity estimates have relatively low variance such that it does not significantly change the reported Pearson correlation coefficients. Hence, they are only computed once.

### D.4 LIPSCHITZ CONSTANT OF THE OPTIMIZED ENTROPY MODEL

The Lipschitz constant of $r \in \mathcal{V}_r$ is approximated using the power iteration method on its Jacobian:

$$\bar{\mathrm{Lip}}(r) = \frac{1}{N} \sum_{\mathbf{y} \in \mathcal{D}} \sqrt{b_K^\top J_r^\top(\mathbf{y}) J_r(\mathbf{y}) b_K}; \quad b_{k+1} = \frac{J_r^\top(\mathbf{y}) J_r(\mathbf{y}) b_k}{\|J_r^\top(\mathbf{y}) J_r(\mathbf{y}) b_k\|_2}, \tag{84}$$

where $b_0$ is initialized randomly such that $\|b_0\|_2 = 1$. Figure 6a shows these measures per split point, for $N = 100, K = 1,000, T = 512$.

Figure 6b shows, for the codecs previously evaluated, the correlation between the Lipschitz constant of the trained entropy models and the bitrate of their target representations. There is an average Pearson correlation of -0.32 between these quantities, across methods. These results suggest that, for transformers using the proposed entropy model, the optimization process favors simpler hypotheses as the split point $S$ is placed deeper.

