# OpenReview forum: "Rate-Distortion Optimization for Transformer Inference"
_ICLR.cc/2026/Conference — Submitted to ICLR 2026_

### Official Review · Reviewer_yFGQ · 2025-10-27

**Soundness:** 3
**Presentation:** 2
**Contribution:** 3
**Rating:** 4
**Confidence:** 4

**Summary:**

The authors address the problem of rate–distortion optimization within the framework of transformer inference. They propose a new entropy model and introduce a theoretical framework based on the v-entropy gap and its bound.

**Strengths:**

The topic is interesting and relevant, and the theoretical analysis has potential.

**Weaknesses:**

The paper suffers from several critical issues that need to be addressed before it can be considered for publication.
- The motivation of the paper is not sufficiently grounded in real-world applications. The practical context in which the proposed problem formulation becomes relevant should be clarified. The authors should provide concrete examples of situations where rate–distortion optimization for transformer inference arises in practice. For instance, this topic appears closely related to federated learning and distributed inference, where lossy compression is often employed to reduce communication costs. The authors are encouraged to make this connection explicit and cite related references to position their contribution within this broader context. Another example: In Section 3.1, the paper states: "We propose a transformer-based entropy model that does not have direct access to elements in Y, requiring more support from the hyper-prior." This point requires further explanation. Why is this assumption made? Does it model a more challenging or more realistic scenario? Clarifying the motivation and implications of this design choice would help readers better understand its practical relevance.
- The connection between the theoretical framework and the experimental observations is currently weak. The authors report interesting empirical findings (e.g., the rate–distortion performance as a function of the split point) and hypothesize that these trends are due to the v-entropy gap and its generalization error. However, no concrete evidence or analysis is provided to substantiate this claim. The statement:"As the source is further processed by non-linear functions in a high-dimensional space, the complexity of the intermediate representations increases, making their probability density more difficult to estimate" is plausible but lacks empirical validation. Can the authors provide any proof or quantitative evidence that the complexity of intermediate representations indeed increases as hypothesized? Moreover, does this increase correlate with the measured rate–distortion performance? Establishing such a link would significantly strengthen the paper’s contribution and credibility.

Minor remarks:
- Related Work: The related work section needs substantial improvement. The authors mention learning-based image compression methods and claim that these approaches primarily rely on convolutional modules. However, many recent image compression architectures are transformer-based. The authors should clarify how their method relates to these transformer-based models and cite relevant works. Another example: the authors claim that, in the framework of image compression, the target representation follows a multivariate normal distribution with diagonal covariance. However, alternative distributions have been explored in the literature. For example, see:
Fu, H., Liang, F., Lin, J., Li, B., Akbari, M., Liang, J., & Han, J. (2023). Learned image compression with gaussian-laplacian-logistic mixture model and concatenated residual modules. IEEE Transactions on Image Processing, 32, 2063–2076.

**Questions:**

N/A

---

> ### Author Response · Authors · 2025-11-21
> **Opening Message**
>
> We appreciate the reviewers for taking their time to analyze the paper and give us feedback. We understand that the nature of our work can be considered niche to the ICLR community. However, we believe that information theory is the cornerstone of representation learning and, as such, ICLR is the best venue for our work.
>
> Please see the additional messages for our responses to each of the reviewers’ comments. We apologize for the several notifications they might receive. Since some of the suggestions/weaknesses are not numbered, we decided to include them in each comment. We have submitted a PDF as supplementary material highlighting the content differences between the submitted version and the rebuttal version. The new content is highlighted in blue, whereas the erased content is highlighted in red.
>
> We hope to have addressed most concerns satisfactorily and will happily participate in further discussion as requested by the reviewers.

---

> > ### Author Response · Authors · 2025-11-21
> >
> > ### The motivation of the paper is not sufficiently grounded in real-world applications. The practical context in which the proposed problem formulation becomes relevant should be clarified. The authors should provide concrete examples of situations where rate–distortion optimization for transformer inference arises in practice. For instance, this topic appears closely related to federated learning and distributed inference, where lossy compression is often employed to reduce communication costs. The authors are encouraged to make this connection explicit and cite related references to position their contribution within this broader context. Another example: In Section 3.1, the paper states: "We propose a transformer-based entropy model that does not have direct access to elements in Y, requiring more support from the hyper-prior." This point requires further explanation. Why is this assumption made? Does it model a more challenging or more realistic scenario? Clarifying the motivation and implications of this design choice would help readers better understand its practical relevance.
> >
> > The Abstract and the first paragraph of Section 1 (Introduction) motivated the problem space by pointing out the high resource requirements of transformers and the need for horizontal scaling across heterogeneous devices. The first paragraph of Section 1 had a concrete example: a mobile phone runs inference on the first few layers of a transformer, efficiently compresses the intermediate representation and transmits the resulting bitstream to a cloud server that decodes the representation and runs inference on the rest of the transformer. This is an example of distributed inference. Figure 1 showed an architecture overview that explicitly presented the problem space and our solution at a high-level. We also pointed out in the paper that it is important to improve the rate-distortion performance as much as possible in order to reduce latency while maintaining task performance.
> >
> > In Section 2 (Previous work), we go over the current work in the area of learnable *coding for machines*, which is the main topic of our work. We mention works such as Choi \& Bajic (2022), Harell et al. (2022), Bajic (2025), de Andrade \& Bajic (2024), and  Harell et al. (2025). All this work is closely related to ours and we point out that while their focus is on computer vision tasks using convolutional neural networks, our work focuses on auto-regressive tasks using transformers, while applying the proposed method to language models.
> >
> > To motivate the design choices made around the entropy model. We added the following to Section 3.1 (An auto-regressive hyper-prior is all you need):
> >
> > > "This design choice allows to code and transmit all representation elements within a time frame in parallel. It simplifies the entropy model of the target representation by assuming full independence of its elements given the hyper-prior. In Section 3.3, we compare our proposed method against a baseline that does not make this design choice, showing relatively better performance. The theory developed in Section 4.2 shows that the complexity of the entropy model increases the generalization error of the $\mathcal{V}$-entropy gap and this can negatively affect the rate-distortion performance of a codec."
> >
> > We also made some allusions to this aspect in Section 5 (Summary and conclusion). Please take a look at the changes in the new version of the paper.

---

> > ### Author Response · Authors · 2025-11-21
> >
> > ### The connection between the theoretical framework and the experimental observations is currently weak. The authors report interesting empirical findings (e.g., the rate–distortion performance as a function of the split point) and hypothesize that these trends are due to the v-entropy gap and its generalization error. However, no concrete evidence or analysis is provided to substantiate this claim. The statement: As the source is further processed by non-linear functions in a high-dimensional space, the complexity of the intermediate representations increases, making their probability density more difficult to estimate" is plausible but lacks empirical validation. Can the authors provide any proof or quantitative evidence that the complexity of intermediate representations indeed increases as hypothesized? Moreover, does this increase correlate with the measured rate–distortion performance? Establishing such a link would significantly strengthen the paper’s contribution and credibility.
> >
> > In Section 4.4 (Rate-distortion performance and the split point), we computed  approximations of the covariance determinant and Rademacher complexity of the target representations across multiple layers, for different models and tasks. These measures correspond to terms present in the bounds given by Theorems 2 and 3, respectively. As stated in that Section, the other terms in these bounds are not computed because they act as constants, since the architecture of the entropy model remains the same across layers.
> >
> > Our results in Section 4.4. and Appendix D (Rate-distortion performance and the split point: additional details and results) show that there is a strong positive correlation between these two measures and the rate attained on the target representations. These results seem to explain the change in rates between split points. We conclude that the $\mathcal{V}$-entropy gap and its generalization error can also increase with specific notions of complexity of the target representation, which, unlike its entropy, can increase in deeper layers.
> >
> > Sections 1 (Introduction), 4 (Experimental results) and 5 (Summary and conclusion) discussed the results and provided some insights. As per the reviewer's suggestion, we have expanded on the intuition and the connection between the theory and the experiments of Section 4.3. Please take a look at the changes in Sections 3.2 and 3.3.
> >
> > ### Related Work: The related work section needs substantial improvement. The authors mention learning-based image compression methods and claim that these approaches primarily rely on convolutional modules. However, many recent image compression architectures are transformer-based. The authors should clarify how their method relates to these transformer-based models and cite relevant works. Another example: the authors claim that, in the framework of image compression, the target representation follows a multivariate normal distribution with diagonal covariance. However, alternative distributions have been explored in the literature. For example, see: Fu, H., Liang, F., Lin, J., Li, B., Akbari, M., Liang, J., \& Han, J. (2023). Learned image compression with gaussian-laplacian-logistic mixture model and concatenated residual modules. IEEE Transactions on Image Processing, 32, 2063–2076.
> >
> > In Section 2 (Previous work), we did cite more recent work that uses transformer architectures (Zou et al., 2022; Li et al., 2024) and even state-space models (Qin et al., 2024; Zeng et al., 2025). We discussed the transformer architectures in some detail. The work in *coding for machines* so far has been limited to convolutional neural networks and computer vision tasks. The use of transformers for computer vision tasks might be presented soon.
> >
> > Assuming a multivariate Gaussian distribution for the target representation is pretty standard in literature. Although the work from Fu et al. (2023) considers a different distribution, recent state-of-the-art work such as Jiang et al. (2025) refers to this work, yet still assumes a Gaussian distribution. We informally believe that, in practice, an expressive enough analysis transform can adapt to either of these distributions. This is probably why we do not see any advantage in using the more expressive Fourier basis approach to model the hyper-prior.

---

> > > ### Comment · Reviewer_yFGQ · 2025-11-25
> > >
> > > Thank you for the responses. Unfortunately, I am not convinced that the revised version improves the paper. I will point out just one example: the manuscript states that “the distribution imposed on the target representation is a multivariate normal distribution,” even though other choices exist, yet none of these alternatives are cited. This is a significant oversight, especially in the related work section, where the goal should be to provide a comprehensive view of existing approaches rather than selectively highlighting those that support your narrative. For these reasons, I will keep my score unchanged.

---

### Official Review · Reviewer_cubh · 2025-10-31

**Soundness:** 3
**Presentation:** 3
**Contribution:** 3
**Rating:** 6
**Confidence:** 2

**Summary:**

The paper proposes a learnable, lossy codec to compress transformer intermediate representations for split-device inference, optimizing an explicit rate–distortion trade-off. Its key design is a simple hyper-prior–centric entropy model: a transformer generates a compact hyper-prior that conditions a second transformer predicting factorized Gaussian parameters for the target activations, without direct access to previously coded targets; a deep factorized CDF models the hyper-prior. The authors introduce the V-entropy gap—the difference between achievable rate under a restricted predictive family and true entropy—show that the codec’s rate term minimizes this gap, and derive bounds linking it to covariance determinant, KL divergence, Lipschitz constants, and provide PAC-style generalization bounds via Rademacher complexity. Empirically, on GPT-2/Pythia (OpenWebText) and ViT/ResNet (ImageNet), the method achieves better RD than a Fourier density hyper-prior and a more complex direct-access entropy model (e.g., ~10.7% BD-rate gain vs direct-access), beats Deflate in bitrate and speed on activations, and can even improve task metrics at early splits. A key finding is that transformer bitrate increases with deeper splits—despite decreasing entropy—strongly correlating with covariance determinant and Rademacher complexity; for ResNet, the trend reverses. This supports a bias–variance-like explanation: increasing model complexity may reduce the gap but harms generalization. Limitations include factorized Gaussian assumptions, approximate bounds, and a non-optimized implementation.

**Strengths:**

1. Introduces the V-entropy gap as a unifying lens to explain why learned codecs operate above entropy in practice, bridging information-theoretic concepts (usable information, conditional V-entropy) with real transformer activation compression. The hyper-prior–centric design that conditions only on W (no direct access to past Y) challenges the necessity of complex autoregressive context models and is well-suited to streaming/split inference.
2. Theory-practice alignment: formal definitions and bounds (covariance-determinant upper bounds, Lipschitz/PAC–Rademacher generalization bounds) with proofs, plus comprehensive experiments across architectures and modalities (GPT-2, Pythia, ViT, ResNet). Empirically outperforms more complex baselines in BD-rate, shows clear bitrate/speed gains over Deflate, and validates depth-dependent rate trends via high correlations with covariance determinant and Rademacher complexity.

**Weaknesses:**

1. Limited impact of the derived bounds on empirical performance: While the covariance- and PAC/Rademacher–Lipschitz bounds are useful for explaining observed trends (e.g., higher bitrate at deeper layers, capacity–generalization trade-offs), they are loose and approximation-heavy, offering little prescriptive guidance for hyperparameters or architecture.

2. Lack of real world evaluation: This paper does not demonstrate end-to-end gains in a true multi-GPU or cross-machine setup (e.g., wall-clock latency/throughput improvements under PCIe/NVLink/Ethernet, contention under batching, packetization/overheads). That is a genuine external validity gap.

**Questions:**

- In Figure 5, the caption "The rate is measured in bits-per-token (BPT) using the preprocessing target resolution of 224 × 224 pixels." is confusing. I suppose it's a figure for text tasks but why is pixels mentioned here?

---

> ### Author Response · Authors · 2025-11-21
> **Opening Message**
>
> We appreciate the reviewers for taking their time to analyze the paper and give us thoughtful feedback. We understand that the nature of our work can be considered niche to the ICLR community. However, we believe that information theory is the cornerstone of representation learning and, as such, ICLR is the best venue for our work.
>
> Please see the additional messages for our responses to each of the reviewers’ comments. We apologize for the several notifications they might receive. Since some of the suggestions/weaknesses are not numbered, we decided to include them in each comment. We have submitted a PDF as supplementary material highlighting the content differences between the submitted version and the rebuttal version. The new content is highlighted in blue, whereas the erased content is highlighted in red.
>
> We hope to have addressed most concerns satisfactorily and will happily participate in further discussion as requested by the reviewers.

---

> > ### Author Response · Authors · 2025-11-21
> >
> > ### Limited impact of the derived bounds on empirical performance: While the covariance- and PAC/Rademacher–Lipschitz bounds are useful for explaining observed trends (e.g., higher bitrate at deeper layers, capacity–generalization trade-offs), they are loose and approximation-heavy, offering little prescriptive guidance for hyperparameters or architecture.
> >
> > One piece of guidance that is provided by the insights of this work is that limiting the representation complexity in transformers can lead to significant rate-distortion gains in deeper layers. We hope to evaluate this idea in future work. To verbalize this, we added the following to Section 5 (Summary and conclusion):
> >
> > > "We observe that only in the ResNet task considered, the covariance determinant and Rademacher complexity of the target representations decrease in deeper layers, along with their rate. We conclude that in transformers, the intermediate representations from deeper layers require an increase in complexity, which, as shown in this work, causes their rate to also increase. This transformer trait seems to be required in order to perform the tasks considered. We hope to develop potential solutions in future work."
> >
> > We believe that although the bounds are approximation-heavy and might be loose, they could still prove useful in reducing the complexity of the target representations, and thus the $\mathcal{V}$-entropy gap.
> >
> > ### Lack of real world evaluation: This paper does not demonstrate end-to-end gains in a true multi-GPU or cross-machine setup (e.g., wall-clock latency/throughput improvements under PCIe/NVLink/Ethernet, contention under batching, packetization/overheads). That is a genuine external validity gap.
> >
> > The speed comparisons highlight the need to optimize our current implementation of the proposed method. In the compression community, researchers often favor rate gains over speed. Once a coding standard has been drafted, the arduous process of optimizing a reference implementation begins. For our proposed method, this could mean simplifying the architecture, approximating computations, optimizing subroutines, and providing hardware support for key components. Only then we could truly assess the end-to-end gains of this approach.
> >
> > We try to motivate the potential gains of the proposed method by pointing out that under the current suboptimal performance, or method has an advantage when the effective link speed is less than 37.77 Mbps. This amount of bandwidth is common in some mobile networks. This estimate considers a conservative communication protocol overhead of 9\%. Since our unoptimized method has a low rate-speed ratio, additional communication latencies, packetization overheads, and contention delays that scale with rate would make a stronger case in favor of our proposed method.
> >
> > We consider our strongest contributions to be theoretical and believe that a better codec for language models, inspired by ours, can be proposed in future work. We hope our current work motivates this type of solution and establishes the necessary background.
> >
> > ### In Figure 5, the caption "The rate is measured in bits-per-token (BPT) using the preprocessing target resolution of 224 × 224 pixels." is confusing. I suppose it's a figure for text tasks but why is pixels mentioned here?
> >
> > Thank you for pointing this out. We apologize for the confusion. That information belonged to a different figure and it is no longer relevant since the rates presented in the correct figure have been normalized. We have corrected the mistake.

---

### Official Review · Reviewer_upUu · 2025-11-01

**Soundness:** 2
**Presentation:** 2
**Contribution:** 2
**Rating:** 2
**Confidence:** 3

**Summary:**

This paper investigates the compression of intermediate features in Transformers and identifies the counterintuitive phenomenon where deeper split points yield higher bitrates. A theoretical explanation is proposed, and preliminary comparative experiments on compression networks are conducted.

**Strengths:**

The paper tries to employs rigorous and detailed theoretical analysis, formalized through theorems.

**Weaknesses:**

* The paper resembles the contributions from three theorems, without explaining how experimental results are connected to these contributions.
* Experimental results are quite limited from the scope of image compression. It is suggested to add comparisons with SOTA methods and better model tuning. For the Hyperprior model, the authors explore factorized priors and Fourier bases, but the performance of the Fourier basis is significantly worse, suggesting potential inadequate tuning. Regarding the encoding of the latent `y`, an autoregressive model is used. While the proposed method performs better on the Perplexity metric, the autoregressive model achieves better results on the LAMBADA benchmark. Furthermore, experiments may include comparisons with state-of-the-art compression methods.
* The information processing inequality suggests that deeper layers should have lower entropy. However, Section 4.4 attempts to argue that additional non-linear mappings push features into higher-dimensional spaces, increasing the complexity of entropy estimation and raising the upper bound of the generalization error.
* Some results are not well-explained. The figures indicate that both Transformer and ViT models exhibit higher bitrates with deeper split points, whereas ResNet does not. This observation does not seem to be adequately explained by the proposed theory.
* Baselines are also incomplete. The lossless compression comparison only uses Deflate; adding methods like zstd or tensor quantization is recommended.
* There are several writing issues. 1) V-entropy Presentation: It is recommended to include Appendix Table 3 and Table 4 for better illustration. 2) Outdated References: Section 2's related work (latest 2018) requires updating. 3) Incorrect Title: Sec. 3.1 title should be "Token-level Causal Hyperprior." 4) Unclear Terminology: "Direct access" and Fig. 2b's "Direct-access entropy model" lack immediate explanation. 5) Notation Error: Theorem 1's Gv(Y|Y) should likely be Gv(Y|W).

**Questions:**

Please refer to the weaknesses.

---

> ### Author Response · Authors · 2025-11-21
> **Opening Message**
>
> We appreciate the reviewers for taking their time to analyze the paper and give us feedback. We understand that the nature of our work can be considered niche to the ICLR community. However, we believe that information theory is the cornerstone of representation learning and, as such, ICLR is the best venue for our work.
>
> Please see the additional messages for our responses to each of the reviewers’ comments. We apologize for the several notifications they might receive. Since some of the suggestions/weaknesses are not numbered, we decided to include them in each comment. We have submitted a PDF as supplementary material highlighting the content differences between the submitted version and the rebuttal version. The new content is highlighted in blue, whereas the erased content is highlighted in red.
>
> We hope to have addressed most concerns satisfactorily and will happily participate in further discussion as requested by the reviewers.

---

> > ### Author Response · Authors · 2025-11-21
> >
> > ### The paper resembles the contributions from three theorems, without explaining how experimental results are connected to these contributions.
> >
> > In Section 3.2 (The $\mathcal{V}$-entropy gap and its bounds), we explained how Theorem 1 is connected to the objective function. In Section 4.3 (Rate-distortion performance and the split point) we measured approximations of the covariance determinant and Rademacher complexity of the target representations, key concepts that appear in the bounds shown by Theorems 2 and 3. We only focused on these concepts because the other terms are fixed across layers. Sections 1 (Introduction), 4 (Experimental results) and 5 (Summary and conclusion) discussed the results and provided some insights.
> >
> > As per the reviewer's suggestion, we have expanded on the intuition and the connection between the theory and the experiments of Section 4.3. Please take a look at the changes in Sections 3.2 and 3.3. If the reviewer if not satisfied with our new explanations, could they please point out more specifically what kind of connection is missing so we can promptly address it?
> >
> > Finally, our work has theoretical contributions that go beyond the three theorems presented in the main sections. We briefly introduce other theoretical findings that are fully presented and discussed in Appendix A.
> >
> > ### Experimental results are quite limited from the scope of image compression. It is suggested to add comparisons with SOTA methods and better model tuning.
> >
> > Our focus is on language models because of their auto-regressive nature. Because the proposed entropy models assign probability distributions auto-regressively across time steps, our method supports auto-regressive predictions. This was summarized in the paper as follows:
> >
> > > "The masks for the attention mechanisms in $h$ and $g_y$ are restricted to create representations that only depend on the current and previously coded elements. Although this restriction is not required to code the hyper-prior or the target representation, it allows to grow the existing side information by only appending elements to it as the time series is further processed. This is a critical feature for auto-regressive tasks that avoids the transmission of an entire hyper-prior for the inference of a new time frame; only the new elements generated by the new time frame need to be transmitted."
> >
> > To evaluate the bounds in Theorems 2 and 3, we included two different models trained on image classification. However, it is not the purpose of those experiments, or this work, to achieve state-of-the-art performance on image reconstruction (compression) tasks. Since our strongest contributions are theoretical, we also do not strive or claim to achieve state-of-the-art performance on the compression of the intermediate representations generated by language models. However, because we are not aware of other methods that attempt this, our method could very well be the best out there, at least for now.
> >
> > Could the reviewer please clarify why, for the purpose of the contributions in this work, is image compression more relevant than the other tasks presented in this work?
> >
> > ### For the Hyperprior model, the authors explore factorized priors and Fourier bases, but the performance of the Fourier basis is significantly worse, suggesting potential inadequate tuning.
> >
> > We believe the results to be accurate. Initial experiments using this approach performed much worse than what was reported. Several hyper-parameters and quantization methods were evaluated. The method has not been extensively applied to codecs so we have no reason to believe it should perform better. The original authors did show that the method successfully adapts to more complex probability distributions, but as we have shown in this work, simplicity can yield better rate-distortion performance.

---

> > ### Author Response · Authors · 2025-11-21
> >
> > ### Regarding the encoding of the latent y, an auto-regressive model is used. While the proposed method performs better on the Perplexity metric, the auto-regressive model achieves better results on the LAMBADA benchmark.
> >
> > We found the LAMBADA benchmark to be very sensitive to text pre-processing and post-processing techniques. Its evaluation dataset is small, so it is easy to tilt the balance in favor of a method by changing these details. We chose the most sensible, yet simple, approach, which indeed shows that the direct-access entropy model performs relatively better at this task.
> >
> > We present the LAMBADA results as a way to corroborate that the resulting language models can perform language tasks reasonably well. We found this was needed because the size of the language models that we evaluated is small, and because we also believe that the cross-entropy classification performance can be a bit deceiving in assessing this aspect. We wanted to know by how much an increase in cross-entropy classification performance can translate into language task performance.
> >
> > We would certainly need more tests to evaluate the overall performance on language tasks. However, in the work we have seen, this performance seems to correlate well with the cross-entropy classification performance.
> >
> > ### Furthermore, experiments may include comparisons with state-of-the-art compression methods.
> >
> > Most task-specific learnable compression methods in literature are developed in the context of image and video compression. These models would need to be adapted to support machine tasks, auto-regressive predictions, and unidimensional data. The direct-access entropy model that we use as a benchmark is an attempt to do just that. It follows the overall architecture of Ballé et al. (2018), which is the basis for most state-of-the-art image codecs (He et al., 2022; Zou et al., 2022; and Jiang
> > et al., 2025). It has also been used in the context of *coding for machines* for diverse computer vision tasks (Choi \& Bajic, 2022; de Andrade \& Bajic, 2024; and Harell et al., 2025).
> >
> > ### The information processing inequality suggests that deeper layers should have lower entropy. However, Section 4.4 attempts to argue that additional non-linear mappings push features into higher-dimensional spaces, increasing the complexity of entropy estimation and raising the upper bound of the generalization error.
> >
> > Indeed, the entropy of an intermediate representation can only decrease as we go deeper, but the rate does not have to correlate with it. Until now, this was often an expectation, but as we have shown in this work, both theoretically and empirically, this is not necessarily the case for transformers. Our contributions contradict the intuition that many researchers in the compression community might have. It certainly surprised us, which led us to develop the proposed theory to understand this behaviour.
> >
> > ### Baselines are also incomplete. The lossless compression comparison only uses Deflate; adding methods like zstd or tensor quantization is recommended.
> >
> > We have added the results from Zstandard (zstd). Please see the changes made to Section 4.2. (Bitrate and speed comparisons).
> >
> > In terms of tensor quantization, our entropy model could be used to dictate the bit allocation of each dimension of a target representation. The quantization levels are grounded in theory, since they would be the product of a rate-distortion optimization. We do not believe that the rate-distortion performance of this technique will be as good as using our proposed entropy model to code the target representations. Since this technique would just be a byproduct of our proposed method, we do not see much value in such analysis.

---

> ### Author Response · Authors · 2025-11-21
>
> ### Some results are not well-explained. The figures indicate that both Transformer and ViT models exhibit higher bitrates with deeper split points, whereas ResNet does not. This observation does not seem to be adequately explained by the proposed theory.
>
> The theory developed still captures this behavior since, even in ResNets, the covariance determinant and the Rademacher complexity of the target representations strongly correlate with rate. Thus, we observe that in ResNets, these complexity measures decrease in deeper layers along with the rate. The conclusion we draw is that transformers require an increase in representation complexity as they go into deeper layers. We do not know what properties in the transformer architecture cause this behavior, but this trait seems to be required in order to perform the machine tasks considered. We could hypothesize several reasons and solutions, but we would rather research them thoroughly in future work. Nevertheless, the theory developed holds in this case, and we consider that further explaining this behavior is outside the scope of this work.
>
> We added the following to Section 5 (Summary and conclusion):
>
> > "We observe that only in the ResNet task considered, the covariance determinant and Rademacher complexity of the target representations decrease in deeper layers, along with their rate. We conclude that in transformers, the intermediate representations from deeper layers require an increase in complexity, which, as shown in this work, causes their rate to also increase. This transformer trait seems to be required in order to perform the tasks considered. We hope to develop potential solutions in future work."
>
> ### There are several writing issues. 1) V-entropy Presentation: It is recommended to include Appendix Table 3 and Table 4 for better illustration. 2) Outdated References: Section 2's related work (latest 2018) requires updating. 3) Incorrect Title: Sec. 3.1 title should be "Token-level Causal Hyperprior." 4) Unclear Terminology: "Direct access" and Fig. 2b's "Direct-access entropy model" lack immediate explanation. 5) Notation Error: Theorem 1's Gv(Y|Y) should likely be Gv(Y|W).
>
> 1) Tables 3 and 4 take a considerable amount of space that we do not have for the main sections. However, their contents are visually summarized by Figure 4.
>
> 2) We added more recent references for image compression models. The work from 2018 is seminal. Most of the other recent work had already been cited in more specific contexts.
>
> 3) The term "token-level" makes allusion to language models. The proposed entropy model can be used for other auto-regressive tasks besides language tasks. Our subtitle is an Easter egg that refers to the "Attention is all you need" seminal paper on transformers.
>
>     One of the main takeaways of our work is that adding more complexity to the entropy model architecture increases the generalization error and can negatively affect performance. We outperform the baselines with a simpler approach, which explains the title.
>
> 4) Thank you for pointing this out. We fixed an incorrect reference to the wrong subfigure, moved the architecture figures higher up, and added a few connections to the "direct access" concept. Otherwise, we believe that the key aspect of the direct-access entropy model is expressed as soon as it is introduced.
>
> 5) We believe that $G_{\mathcal{V}}(Y|Y)$ is correct.

---

### Official Review · Reviewer_1DH4 · 2025-11-06

**Soundness:** 2
**Presentation:** 2
**Contribution:** 1
**Rating:** 0
**Confidence:** 4

**Summary:**

The authors develop a lossy compression algorithm for encoding neural network outputs. They achieve this by selecting an appropriate hidden layer in the network architecture and quantising and entropy coding the network activations at that layer. To improve the efficiency of their codec, the authors propose encoding side information, for which they develop a hyperprior model.

They also develop a PAC bound to study the rate-distortion tradeoff of their method. Finally, they conduct experiments on compressing large language model (LLM) outputs.

**Strengths:**

I found the experiments on the performance of the split points, i.e., which hidden layer to pick and quantise, interesting.

**Weaknesses:**

The paper's most significant weakness is its lack of both novelty and insight.

The core idea of the paper, as I mentioned in the summary, is to encode neural network outputs by quantising and entropy coding hidden layer activations. This idea, along with the use of a hyperprior model, is not a novel or significant contribution.

Furthermore, the authors' theoretical results follow easily from basic information-theoretic identities. This in itself is not an issue, but the final bound the authors obtain in Thm 3 is not directly useful: the authors do not compute it numerically, so it is difficult to judge its tightness, but based on my knowledge of similar bounds, I would imagine it is hopelessly loose. Indeed, it is not entirely clear what the authors' intention was in including the entire theoretical discussion in Sections 3.2 and 3.3, as these results do not yield any actionable insights.

Furthermore, in certain places, the authors' language is highly imprecise, even contradictory, and potentially misleading. The main offenders are:
 - The authors refer to "differentiable quantization functions." -- No quantization function is differentiable. To someone who doesn't know the works the authors cite, this is extremely misleading terminology. Something like "quantizer with a differentiable training-time approximation" would be appropriate.
 - "This quantization (rounding) function discretizes the target representation so it can be coded and so gradients can flow through it during automatic differentiation" -- this sentence is contradictory, a function with a discrete range cannot be differentiable.
 - "Deflate operates at 0.672 milliseconds per token, whereas our proposed method operates at 0.348 milliseconds per token" -- once again, this is a highly misleading sentence. In Appendix C, the authors state that: "The hyper-prior and entropy model ran on an NVIDIA 2080 RTX Ti GPU. The arithmetic coder ran on a single core of an Intel Core i9-9900K CPU. The Deflate algorithm ran on the same system using a single CPU core." Unless the authors can demonstrate that the compute required to run the hyperprior model is insignificant compared to AC, the comparison of their method to Deflate is unfair.

More miscellaneous points:
 - The AE and AD acronyms in Figure 1 are not defined.
 - Eq 7: The notation $ (g[f(x)] \circ f) (x)$ is really hard to read; the notation $g[\cdot]$ is undefined, my assumption for reading was that $g[x] = g_x$.
 - Eq 8: $r \in \{ r \mid r(y)=−\log g(y), g\in V\}$ using $r$ for both the bound variable and a generic element of the set.
 - Accuracy is not a good y-axis label in Figure 3, since lower values are better in their plots.
 - The authors don't specify what distortion they used in their experiments; I presume they used the standard cross-entropy loss.

**Questions:**

n/a

---

> ### Author Response · Authors · 2025-11-21
> **Opening Message**
>
> We appreciate the reviewers for taking their time to analyze the paper and give us feedback. We understand that the nature of our work can be considered niche to the ICLR community. However, we believe that information theory is the cornerstone of representation learning and, as such, ICLR is the best venue for our work.
>
> Please see the additional messages for our responses to each of the reviewers’ comments. We apologize for the several notifications they might receive. Since some of the suggestions/weaknesses are not numbered, we decided to include them in each comment. We have submitted a PDF as supplementary material highlighting the content differences between the submitted version and the rebuttal version. The new content is highlighted in blue, whereas the erased content is highlighted in red.
>
> We hope to have addressed most concerns satisfactorily and will happily
> participate in further discussion as requested by the reviewers.

---

> > ### Author Response · Authors · 2025-11-21
> >
> > ### This idea, along with the use of a hyperprior model, is not a novel or significant contribution.
> >
> > As far as we know, learnable codecs for rate-distortion optimization have not been used in the context of auto-regressive tasks using transformers, yet alone for language modeling.
> >
> > The proposed codec is unique in that the hyper-prior is the only context used to make predictions of the target representation. Previous approaches that use an hyper-prior usually assign a small amount of information to it, since they rely on previously coded elements of the target representation as context. Using these previous approaches, we have informally seen that the usage of a hyper-prior does not grant significant rate reductions in some situations.
> >
> > The baselines proposed for comparison against our proposed model are also novel. They do not perform as well in our particular application, but could be viable in other tasks if they exploit their inductive biases. We could not use as baselines the existing methods we know, without adaptation.
> >
> > Our strongest contributions are theoretical. Using the proposed method and its baselines, we consistently noticed a decrease in rate-distortion performance as we code representations from deeper layers. This behavior would be considered unexpected (but still possible) by most people in the compression community. We are able to identify that as we go deeper into the transformer architecture, increases in the complexity of the target representation push the rate higher. We show this theoretically and empirically.
> >
> > Overall, we extend the existing theory around usable information to measure the difference between rate and entropy, and establish the generalization error of this concept. We derive several bounds for these concepts, offering several insights along the way.
> >
> > ### The authors' theoretical results follow easily from basic information-theoretic identities. This in itself is not an issue, but the final bound the authors obtain in Thm 3 is not directly useful: the authors do not compute it numerically, so it is difficult to judge its tightness, but based on my knowledge of similar bounds, I would imagine it is hopelessly loose. Indeed, it is not entirely clear what the authors' intention was in including the entire theoretical discussion in Sections 3.2 and 3.3, as these results do not yield any actionable insights.
> >
> > In Section 4.4 (Rate-distortion performance and the split point), we computed  approximations of the covariance determinant and Rademacher complexity of the target representations across multiple layers, for different models and tasks. These measures correspond to terms present in the bounds given by Theorems 2 and 3, respectively. As stated in this Section, the other terms in these bounds are not computed because they are constants, since the architecture of the entropy model remains the same across layers.
> >
> > Our results in Section 4.4. and Appendix D (Rate-distortion performance and the split point: additional details and results) show that there is a strong positive correlation between these two measures and the rate attained on the target representations. Although the upper bounds might still be loose, these results seem to explain the change in rates between split points. Computing an approximation of the bounds is achievable, but the quantities they bound — the $\mathcal{V}$-entropy gap and its generalization error — are not easy to compute since the entropy of the target representations is unknown, and the best way to approximate it is the rate itself. As a reminder, we define the $\mathcal{V}$-entropy gap as the difference between rate and entropy.
> >
> > We conclude that the $\mathcal{V}$-entropy gap and its generalization error can also increase with these notions of complexity of the target representation, which, unlike its entropy, can increase in deeper layers. We have expanded on the intuition and the connection between the theory and the experiments of Section 4.3. Please take a look at the changes in Sections 3.2 and 3.3.
> >
> > Other theoretical results are introduced in Sections 3.2 and 3.3, they are fully presented and discussed in Appendix A. We believe that although there are no experimental results attached to these results, and some do not apply to our specific architecture, they still hold value and offer some insights in other and more general settings.

---

> ### Author Response · Authors · 2025-11-21
>
> ### The authors refer to "differentiable quantization functions." -- No quantization function is differentiable. To someone who doesn't know the works the authors cite, this is extremely misleading terminology. Something like "quantizer with a differentiable training-time approximation" would be appropriate.
>
> Thank you for pointing this out. We have made changes accordingly. Please, take a look at the corresponding changes. They should be relatively easy to find in the supplemental document highlighting the differences.
>
>
> ### "This quantization (rounding) function discretizes the target representation so it can be coded and so gradients can flow through it during automatic differentiation" -- this sentence is contradictory, a function with a discrete range cannot be differentiable.
>
> Thank you for pointing this out. Please see the response made above to a similar concern.
>
> ### "Deflate operates at 0.672 milliseconds per token, whereas our proposed method operates at 0.348 milliseconds per token" -- once again, this is a highly misleading sentence. In Appendix C, the authors state that: "The hyper-prior and entropy model ran on an NVIDIA 2080 RTX Ti GPU. The arithmetic coder ran on a single core of an Intel Core i9-9900K CPU. The Deflate algorithm ran on the same system using a single CPU core." Unless the authors can demonstrate that the compute required to run the hyperprior model is insignificant compared to AC, the comparison of their method to Deflate is unfair.
>
> We have now specified in the main Sections of the document that we use a GPU to run the entropy models. We also compared our proposed method against Zstandard. Please take a look at the corresponding changes.
>
> Zstandard does not perform as well as Zlib (DEFLATE) in terms of rate but it is an order of magnitude faster than our implementation of the proposed method.
>
> This section is meant to entertain the idea that the rate gains achieved by our proposed method can justify its additional intrinsic complexity, if any. It also shows that, when the communication bandwidth is relatively low, it is more efficient to compress the intermediate representation than transmitting it as-is. This analysis validates the use of learnable codecs for distributed inference.
>
> The speed benchmarks highlight the need to optimize our implementation. In the compression community, researchers often favor rate gains over speed. Once a coding standard has been drafted, the arduous process of optimizing a reference implementation begins. For our proposed method, this could mean simplifying the architecture, approximating computations, optimizing subroutines, and providing hardware support for key components. We believe that in future work a better codec for language models (based on ours) could be proposed and we hope our current work motivates this type of approach and establishes the necessary background.
>
> ### The AE and AD acronyms in Figure 1 are not defined
>
> The acronyms were defined in the figure caption. We added a brief explanation as to what they do.
>
> ### Eq 7: The notation is really hard to read; the notation $g[\\cdot]$ is undefined, my assumption for reading was that $g[x] = g_x$.
>
> $g[ \\cdot] ( \\cdot ) $ is defined by $\\Omega$ in Section 2.1 (Preliminaries). It is a function that takes a random variable and produces a PMF. The term $(g [ f(\\mathbf{x}) ] \\circ f )(\\mathbf{x})$ in Eq. 7 can be expressed as $g(f(\\mathbf{x}); f(\\mathbf{x}))$: it outputs the probability mass evaluated at $f(\\mathbf{x})$, where the underlying PMF is parameterized by $f(\\mathbf{x})$. We use the $g[ \\cdot] ( \\cdot ) $ notation as opposed to $g(\\cdot;\\cdot)$ to conform to the notation used in the work of Xu et al. (2020), which we rely heavily on.
>
> ### Eq 8: using $r$ for both the bound variable and a generic element of the set.
>
> Thank you for pointing this out. We fixed it in the new version.
>
> ### Accuracy is not a good y-axis label in Figure 3, since lower values are better in their plots.
>
> Thank you for pointing this out. We changed the label to "Distortion".
>
> ### The authors don't specify what distortion they used in their experiments; I presume they used the standard cross-entropy loss.
>
> That is correct. We updated Table 5 (Hyper-parameter settings) to show the distortion functions used in all experiment types.

---

### Meta-Review · Area_Chair_uTh7 · 2026-01-05

**Summary:**

In this work the majority of the reviewers recommend rejection, with the exception of one reviewer who argues for acceptance (though only marginally so and with low confidence).

Issues raised by the reviewers include a lack of novelty and/or insufficient motivation and placement of the work within the literature, limited experimental validation and/or invalid baselines, and some issues with the clarity of the presentation.

In their rebuttal, the authors have made a strong effort to provide responses to many of these issues, but unfortunately I am not convinced that the reviewers' opinions/scores would be significantly changed following the rebuttal.  The somewhat unanimous scoring in favor of rejection from the initial reviews suggests that the work would perhaps be best served by a resubmission to a future venue where the presentation of the work can be modified to include the reviewers' feedback.

**Reviewer Concerns:**

In terms of the novelty of the method and how it compares with other work in the literature, this appears to be largely addressed to me. While some of the reviewer comments about similar ideas existing in the literature are likely valid, the application and auto-regressive setting considered here is novel as far as I am aware.

For the experimental evaluation, the authors have provided some additional baseline comparisons as well as further explanation regarding what each experiment is attempting to demonstrate.  However, given that several reviewers raised concerns regarding the experimental evaluation, it is unfortunately not clear to me that the reviewers would be satisfied with the updated evaluation.  In particular, the authors note that some of the requested baselines are developed for tasks such as image/video compression and would be challenging to adapt to auto-regressive models.  While this is a valid point and perhaps points to the need for methods that address auto-regressive models, as the current work does, it still leaves significant questions as to whether the reviewers would be satisfied with the experimental evaluation/baselines.

In terms of presentation, the authors have provided a revised version of their paper and clarified many of the presentation questions raised by the reviewers.  Yet, given that 3 of the 4 reviewers raise concerns about various aspects of the presentation of the material, it is again unclear whether the reviewers would be satisfied with the revisions, and indeed, the one reviewer who replied post-rebuttal mentions not being satisfied with the revision.

**Reviewer Scores:**

Unfortunately, while the authors have made a strong effort in their rebuttal, given the rather significant distance the scores would need to move I do not believe the scores would be revised significantly enough for the reviewers to recommend acceptance.  One reviewer explicitly mentions not being swayed by the rebuttal/revision, and I suspect all the other reviewers would likely only make relatively minor adjustments to their scores if any.

---

### Decision · Program_Chairs · 2026-01-26

Reject